# WASSERSTEIN EMBEDDING FOR GRAPH LEARNING

**Soheil Kolouri**[*][†]**, Navid Naderializadeh**[*][†]**, Gustavo K. Rohde**[‡]**, & Heiko Hoffmann**[†]

[†]HRL Laboratories, LLC., [‡]University of Virginia

{skolouri,nnaderializadeh,hhoffmann}@hrl.com, gustavo@virginia.edu

## ABSTRACT

We present Wasserstein Embedding for Graph Learning (WEGL), a novel and fast framework for embedding entire graphs in a vector space, in which various machine learning models are applicable for graph-level prediction tasks. We leverage new insights on defining similarity between graphs as a function of the similarity between their node embedding distributions. Specifically, we use the Wasserstein distance to measure the dissimilarity between node embeddings of different graphs. Unlike prior work, we avoid pairwise calculation of distances between graphs and reduce the computational complexity from quadratic to linear in the number of graphs. WEGL calculates Monge maps from a reference distribution to each node embedding and, based on these maps, creates a fixed-sized vector representation of the graph. We evaluate our new graph embedding approach on various benchmark graph-property prediction tasks, showing state-of-the-art classification performance while having superior computational efficiency. The code is available at https://github.com/navid-naderi/WEGL.

## 1 INTRODUCTION

Many exciting and practical machine learning applications involve learning from graph-structured data. While images, videos, and temporal signals (e.g., audio or biometrics) are instances of data that are supported on grid-like structures, data in social networks, cyber-physical systems, communication networks, chemistry, and bioinformatics often live on irregular structures (Backstrom & Leskovec, 2011; Sadreazami et al., 2017; Jin et al., 2017; Agrawal et al., 2018; Naderializadeh et al., 2020). One can represent such data as (attributed) graphs, which are universal data structures. Efficient and generalizable learning from graph-structured data opens the door to a vast number of applications, which were beyond the reach of classic machine learning (ML) and, more specifically, deep learning (DL) algorithms.

Analyzing graph-structured data has received significant attention from the ML, network science, and signal processing communities over the past few years. On the one hand, there has been a rush toward extending the success of deep neural networks to graph-structured data, which has led to a variety of graph neural network (GNN) architectures. On the other hand, the research on kernel approaches (Gärtner et al., 2003), perhaps most notably the random walk kernel (Kashima et al., 2003) and the Weisfeiler-Lehman (WL) kernel (Shervashidze et al., 2011; Rieck et al., 2019; Morris et al., 2019; 2020), remains an active field of study and the methods developed therein provide competitive performance in various graph representation tasks (see the recent survey by Kriege et al. (2020)).

To learn graph representations, GNN-based frameworks make use of three generic modules, which provide i) feature aggregation, ii) graph pooling (i.e., readout), and iii) classification (Hu* et al., 2020). The feature aggregator provides a vector representation for each node of the graph, referred to as a node embedding. The graph pooling module creates a representation for the graph from its node embeddings, whose dimensionality is fixed regardless of the underlying graph size, and which can then be analyzed using a downstream classifier of choice. On the graph kernel side, one leverages a kernel to measure the similarities between pairs of graphs, and uses conventional kernel methods to perform learning on a set of graphs (Hofmann et al., 2008). A recent example of such methods is the framework provided by Togninalli et al. (2019), in which the authors propose a novel node embedding inspired by the WL kernel, and combine the resulting node embeddings with the

---

[*]Denotes equal contribution.

Wasserstein distance (Villani, 2008; Kolouri et al., 2017) to measure the dissimilarity between two graphs. Afterwards, they leverage conventional kernel methods based on the pairwise-measured dissimilarities to perform learning on graphs.

Considering the ever-increasing scale of graph datasets, which may contain tens of thousands of graphs or millions to billions of nodes per graph, the issue of scalability and algorithmic efficiency becomes of vital importance for graph learning methods (Hernandez & Brown, 2020; Hu et al., 2020). However, both of the aforementioned paradigms of GNNs and kernel methods suffer in this sense. On the GNN side, acceleration of the training procedure is challenging and scales poorly as the graph size grows (Bojchevski et al., 2019). On the graph kernel side, the need for calculating the matrix of all pairwise similarities can be a burden in datasets with a large number of graphs, especially if calculating the similarity between each pair of graphs is computationally expensive. For instance, in the method proposed in (Togninalli et al., 2019), the computational complexity of each calculation of the Wasserstein distance is cubic in the number of nodes (or linearithmic for the entropy-regularized distance).

To overcome these issues, inspired by the linear optimal transport framework of (Wang et al., 2013), we propose a linear Wasserstein Embedding for Graph Learning, which we refer to as WEGL. Our proposed approach embeds a graph into a Hilbert space, where the $\ell_2$ distance between two embedded graphs provides a true metric between the graphs that approximates their 2-Wasserstein distance. For a set of $M$ graphs, the proposed method provides:

1. Reduced computational complexity of estimating the graph Wasserstein distance (Togninalli et al., 2019) for a dataset of $M$ graphs from a quadratic complexity in the number of graphs, i.e., $\frac{M(M-1)}{2}$ calculations, to linear complexity, i.e., $M$ calculations of the Wasserstein distance; and

2. An explicit Hilbertian embedding for graphs, which is not restricted to kernel methods, and therefore can be used in conjunction with any downstream classification framework.

We show that compared to multiple GNN and graph kernel baselines, WEGL achieves either state-of-the-art or competitive results on benchmark graph-level classification tasks, including classical graph classification datasets (Kersting et al., 2020) and the recent molecular property-prediction benchmarks (Hu et al., 2020). We also compare the algorithmic efficiency of WEGL with two baseline GNN and graph kernel methods and demonstrate that it is much more computationally efficient relative to those algorithms.

## 2 BACKGROUND AND RELATED WORK

In this section, we provide a brief background on different methods for deriving representations for graphs and an overview on Wasserstein distances by reviewing the related work in the literature.

### 2.1 GRAPH REPRESENTATION METHODS

Let $G = (\mathcal{V}, \mathcal{E})$ denote a graph, comprising a set of nodes $\mathcal{V}$ and a set of edges $\mathcal{E} \subseteq \mathcal{V}^2$, where two nodes $u, v \in \mathcal{V}$ are connected to each other if and only if $(u, v) \in \mathcal{E}$.[1] For each node $v \in \mathcal{V}$, we define its set of neighbors as $\mathcal{N}_v \triangleq \{u \in \mathcal{V} : (u, v) \in \mathcal{E}\}$. The nodes of the graph $G$ may have categorical labels and/or continuous attribute vectors. We use a unified notation of $x_v \in \mathbb{R}^F$ to denote the label and/or attribute vector of node $v \in \mathcal{V}$, where $F$ denotes the node feature dimensionality. Moreover, we use $w_{uv} \in \mathbb{R}^E$ to denote the edge feature vector for any edge $(u, v) \in \mathcal{E}$, where $E$ denotes the edge feature dimensionality. Node and edge features may be present depending on the graph dataset under consideration.

To learn graph properties from the graph structure and its node/edge features, one can use a function $\psi : \mathcal{G} \to \mathcal{H}$ to map any graph $G$ in the space of all possible graphs $\mathcal{G}$ to an *embedding* $\psi(G)$ in a Hilbert space $\mathcal{H}$. Kernel methods have been among the most popular ways of creating such graph embeddings. A graph kernel is defined as a function $k : \mathcal{G}^2 \to \mathbb{R}$, where for two graphs $G$ and $G'$, $k(G, G')$ represents the inner product of the embeddings $\psi(G)$ and $\psi(G')$ over the Hilbert space $\mathcal{H}$. The mapping $\psi$ could be explicit, as in graph convolutional neural networks, or implicit as in the case

---

[1]Note that this definition includes both directed and undirected graphs, where in the latter case, for each edge $(u, v) \in \mathcal{E}$, the reverse edge $(v, u)$ is also included in $\mathcal{E}$.

of the kernel similarity function $k(\cdot, \cdot)$ (i.e., the kernel trick). Kriege et al. (2014) provide a thorough discussion on explicit and implicit embeddings for learning from graphs.

Kashima et al. (2003) introduced graph kernels based on random walks on labeled graphs. Subsequently, shortest-path kernels were introduced in (Borgwardt & Kriegel, 2005). These works have been followed by graphlet and Weisfeiler-Lehman subtree kernel methods (Shervashidze et al., 2009; 2011; Morris et al., 2017). More recently, kernel methods using spectral approaches (Kondor & Pan, 2016), assignment-based approaches (Kriege et al., 2016; Nikolentzos et al., 2017), and graph decomposition algorithms (Nikolentzos et al., 2018) have also been proposed in the literature.

Despite being successful for many years, kernel methods often fail to leverage the explicit continuous features that are provided for the graph nodes and/or edges, making them less adaptable to the underlying data distribution. To alleviate these issues, and thanks in part to the prominent success of deep learning in many domains, including computer vision and natural language processing, techniques based on *graph neural networks (GNNs)* have emerged as an alternative paradigm for learning representations from graph-based data. In general, a GNN comprises multiple hidden layers, where at each layer, each node combines the features of its neighboring nodes in the graph to derive a new feature vector. At the GNN output, the feature vectors of all nodes are aggregated using a readout function (such as global average pooling), resulting in the final graph embedding $\psi(G)$. More details on the combining and readout mechanisms of GNNs are provided in Appendix A.4.

Kipf and Welling (Kipf & Welling, 2016) proposed a GNN architecture based on a graph convolutional network (GCN) framework. This work, alongside other notable works on geometric deep learning (Defferrard et al., 2016), initiated a surge of interest in GNN architectures, which has led to several architectures, including the Graph Attention network (GAT) (Veličković et al., 2017), Graph SAmple and aggreGatE (GraphSAGE) (Hamilton et al., 2017), and the Graph Isomorphism Network (GIN) (Xu et al., 2019). Each of these architectures modifies the GNN combining and readout functions and demonstrates state-of-the-art performance in a variety of graph representation learning tasks.

## 2.2 WASSERSTEIN DISTANCES

Let $\mu_i$ denote a Borel probability measure with finite $p^{\text{th}}$ moment defined on $\mathcal{Z} \subseteq \mathbb{R}^d$, with corresponding probability density function $p_i$, i.e., $d\mu_i(z) = p_i(z)dz$. The 2-Wasserstein distance between $\mu_i$ and $\mu_j$ defined on $\mathcal{Z}, \mathcal{Z}' \subseteq \mathbb{R}^d$ is the solution to the optimal mass transportation problem with $\ell_2$ transport cost (Villani, 2008):

$$\mathcal{W}_2(\mu_i, \mu_j) = \left( \inf_{\gamma \in \Gamma(\mu_i, \mu_j)} \int_{\mathcal{Z} \times \mathcal{Z}'} \|z - z'\|^2 d\gamma(z, z') \right)^{\frac{1}{2}}, \quad (1)$$

where $\Gamma(\mu_i, \mu_j)$ is the set of all transportation plans $\gamma \in \Gamma(\mu_i, \mu_j)$ such that $\gamma(A \times \mathcal{Z}') = \mu_i(A)$ and $\gamma(\mathcal{Z} \times B) = \mu_j(B)$ for any Borel subsets $A \subseteq \mathcal{Z}$ and $B \subseteq \mathcal{Z}'$. Due to Brenier's theorem (Brenier, 1991), for absolutely continuous probability measures $\mu_i$ and $\mu_j$ (with respect to the Lebesgue measure), the 2-Wasserstein distance can be equivalently obtained from

$$\mathcal{W}_2(\mu_i, \mu_j) = \left( \inf_{f \in MP(\mu_i, \mu_j)} \int_{\mathcal{Z}} \|z - f(z)\|^2 d\mu_i(z) \right)^{\frac{1}{2}}, \quad (2)$$

where $MP(\mu_i, \mu_j) = \{f : \mathcal{Z} \to \mathcal{Z}' \mid f_\# \mu_i = \mu_j\}$ and $f_\# \mu_i$ represents the pushforward of measure $\mu_i$, characterized as

$$\int_B d\mu_j(z') = \int_{f^{-1}(B)} d\mu_i(z) \quad \text{for any Borel subset } B \subseteq \mathcal{Z}'. \quad (3)$$

The mapping $f$ is referred to as a transport map (Kolouri et al., 2017), and the optimal transport map is called the Monge map. For absolutely continuous measures, the differential form of the above equation takes the following form, $det(Df(z))p_j(f(z)) = p_i(z)$, which is referred to as the Jacobian equation. For discrete probability measures, when the transport plan $\gamma$ is a deterministic optimal coupling, such a transport plan is referred to as a Monge coupling (Villani, 2008).

Recently, Wasserstein distances have been used for representation learning on graphs and images (Togninalli et al., 2019; Zhang et al., 2020; Bécigneul et al., 2020). In particular, (Togninalli et al., 2019) proposed a Wasserstein kernel for graphs that involves pairwise calculation of the Wasserstein distance between graph representations. Pairwise calculation of the Wasserstein distance, however,

could be expensive, especially for large graph datasets. In what follows, we apply the linear optimal transportation framework (Wang et al., 2013) to define a Hilbertian embedding, in which the $\ell_2$ distance provides a true metric between the probability measures that approximates $\mathcal{W}_2$. We show that in a dataset containing $M$ graphs, this framework reduces the computational complexity from calculating $\frac{M(M-1)}{2}$ linear programs to $M$.

## 3 LINEAR WASSERSTEIN EMBEDDING

Wang et al. (2013) and the follow-up works (Seguy & Cuturi, 2015; Kolouri et al., 2016; Courty et al., 2018) describe frameworks for isometric Hilbertian embedding of probability measures such that the Euclidean distance between the embedded images approximates $\mathcal{W}_2$. We leverage the prior work and introduce the concept of linear Wasserstein embedding for learning graph embeddings.

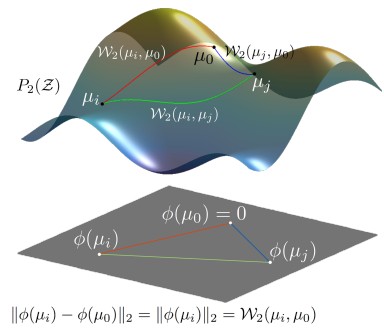

$$\|\phi(\mu_i) - \phi(\mu_0)\|_2 = \|\phi(\mu_i)\|_2 = \mathcal{W}_2(\mu_i, \mu_0)$$
$$\|\phi(\mu_i) - \phi(\mu_j)\|_2 \approx \mathcal{W}_2(\mu_i, \mu_j)$$

Figure 1: Graphical representation of the linear Wasserstein embedding framework, where the probability distributions are mapped to the tangent space with respect to a fixed reference distribution. The figure is adapted from Kolouri et al. (2017).

### 3.1 THEORETICAL FOUNDATION

We adhere to the definition of the linear Wasserstein embedding for continuous measures. However, all derivations hold for discrete measures as well. More precisely, let $\mu_0$ be a reference probability measure defined on $\mathcal{Z} \subseteq \mathbb{R}^d$, with a positive probability density function $p_0$, s.t. $d\mu_0(z) = p_0(z)dz$ and $p_0(z) > 0$ for $\forall z \in \mathcal{Z}$. Let $f_i$ denote the Monge map that pushes $\mu_0$ into $\mu_i$, i.e.,

$$f_i = \mathrm{argmin}_{f \in MP(\mu_0, \mu_i)} \int_{\mathcal{Z}} \|z - f(z)\|^2 d\mu_0(z). \quad (4)$$

Define $\phi(\mu_i) \triangleq (f_i - id)\sqrt{p_0}$, where $id(z) = z, \forall z \in \mathcal{Z}$ is the identity function. In cartography, such a mapping is known as the equidistant azimuthal projection, while in differential geometry, it is called the logarithmic map. The mapping $\phi(\cdot)$ has the following characteristics (partially illustrated in Figure 1):

1. $\phi(\cdot)$ provides an isometric embedding for probability measures, i.e., using the Jacobian equation $p_i = det(Df_i^{-1})p_0(f_i^{-1})$, where $f_i = \frac{\phi(\mu_i)}{\sqrt{p_0}} + id$.

2. $\phi(\mu_0) = 0$, i.e., the reference is mapped to zero.

3. $\|\phi(\mu_i) - \phi(\mu_0)\|_2 = \|\phi(\mu_i)\|_2 = \mathcal{W}_2(\mu_i, \mu_0)$, i.e., the mapping preserves distances to $\mu_0$.

4. $\|\phi(\mu_i) - \phi(\mu_j)\|_2 \approx \mathcal{W}_2(\mu_i, \mu_j)$, i.e., the $\ell_2$ distance between $\phi(\mu_i)$ and $\phi(\mu_j)$, while being a true metric between $\mu_i$ and $\mu_j$, is an approximation of $\mathcal{W}_2(\mu_i, \mu_j)$.

Embedding probability measures $\{\mu_i\}_{i=1}^M$ via $\phi(\cdot)$ requires calculating $M$ Monge maps. The fourth characteristic above states that $\phi(\cdot)$ provides a linear embedding for the probability measures. Therefore, we call it the linear Wasserstein embedding. The mapping $\phi(\mu_i)$ could be thought as the Reproducing Kernel Hilbert Space (RKHS) embedding of the measure, $\mu_i$ (Muandet et al., 2017). In practice, for discrete distributions, the Monge coupling is used, which could be approximated from the Kantorovich plan (i.e., the transport plan) via the so-called barycenteric projection (Wang et al., 2013). We here acknowledge the concurrent work by Mialon et al. (2021), where the authors use a similar idea to the linear Wasserstein embedding as a pooling operator for learning from *sets* of features. The authors demonstrate the relationship between their proposed optimal transport-based pooling operation and the widespread attention pooling methods in the literature. A detailed description of the capabilities of the linear Wasserstein embedding framework is included in Appendix A.1. We next provide the numerical details of the barycenteric projection (Ambrosio et al., 2008; Wang et al., 2013).

## 3.2 NUMERICAL DETAILS

Consider a set of probability distributions $\{p_i\}_{i=1}^M$, and let $Z_i = \left[z_1^i, \ldots, z_{N_i}^i\right]^T \in \mathbb{R}^{N_i \times d}$ be an array containing $N_i$ i.i.d. samples from distribution $p_i$, i.e., $z_k^i \in \mathbb{R}^d \sim p_i, \forall k \in \{1, \ldots, N_i\}$. Let us define $p_0$ to be a reference distribution, with $Z_0 = \left[z_1^0, \ldots, z_N^0\right]^T \in \mathbb{R}^{N \times d}$, where $N = \lfloor \frac{1}{M} \sum_{i=1}^M N_i \rfloor$ and $z_j^0 \in \mathbb{R}^d \sim p_0, \forall j \in \{1, \ldots, N\}$. The optimal transport plan between $p_i$ and $p_0$, denoted by $\pi_i^* \in \mathbb{R}^{N \times N_i}$, is the solution to the following linear program,

$$\pi_i^* = \operatorname{argmin}_{\pi \in \Pi_i} \sum_{j=1}^N \sum_{k=1}^{N_i} \pi_{jk} \|z_j^0 - z_k^i\|^2, \tag{5}$$

where $\Pi_i \triangleq \left\{ \pi \in \mathbb{R}^{N \times N_i} \mid N_i \sum_{j=1}^N \pi_{jk} = N \sum_{k=1}^{N_i} \pi_{jk} = 1, \ \forall k \in \{1, \ldots, N_i\}, \forall j \in \{1, \ldots, N\} \right\}$. The Monge map is then approximated from the optimal transport plan by barycentric projection via

$$F_i = N(\pi_i^* Z_i) \in \mathbb{R}^{N \times d}. \tag{6}$$

Note that the transport plan $\pi_i$ could split the mass in $z_j^0 \in Z_0$ and distribute it on $z_k^i$s. The barycentric projection calculates the center of mass of the transportation locations for $z_j^0$ to ensure that no mass splitting is happening (see Figure 5), and hence it approximates a Monge coupling. Finally, the embedding can be calculated by $\phi(Z_i) = (F_i - Z_0)/\sqrt{N} \in \mathbb{R}^{N \times d}$. With a slight abuse of notation, we use $\phi(p_i)$ and $\phi(Z_i)$ interchangeably throughout the paper. Due to the barycenteric projection, here, $\phi(\cdot)$ is only pseudo-invertible.

## 4 WEGL: A LINEAR WASSERSTEIN EMBEDDING FOR GRAPHS

The application of the optimal transport problem to graphs is multifaceted. For instance, some works focus on solving the "structured" optimal transport concerning an optimal probability flow, where the transport cost comes from distances on an often unchanging underlying graph (Léonard et al., 2016; Essid & Solomon, 2018; Titouan et al., 2019). Here, we are interested in applying optimal transport to measure the dissimilarity between two graphs (Maretic et al., 2019; Togninalli et al., 2019; Dong & Sawin, 2020). Our work significantly differs from (Maretic et al., 2019; Dong & Sawin, 2020), which measure the dissimilarity between non-attributed graphs based on distributions defined by their Laplacian spectra and is closer to (Togninalli et al., 2019).

Our proposed graph embedding framework, termed Wasserstein Embedding for Graph Learning (WEGL), combines node embedding methods for graphs with the linear Wasserstein embedding explained in Section 3. More precisely, let $\{G_i = (\mathcal{V}_i, \mathcal{E}_i)\}_{i=1}^M$ denote a set of $M$ individual graphs, each with a set of possible node features $\{x_v\}_{v \in \mathcal{V}_i}$ and a set of possible edge features $\{w_{uv}\}_{(u,v) \in \mathcal{E}_i}$. Let $h(\cdot)$ be an arbitrary node embedding process, where $h(G_i) = Z_i = \left[z_1, \ldots, z_{|\mathcal{V}_i|}\right]^T \in \mathbb{R}^{|\mathcal{V}_i| \times d}$. Having the node embeddings $\{Z_i\}_{i=1}^M$, we can then calculate a reference node embedding $Z_0$ (see Section 4.2 for details), which leads to the linear Wasserstein embedding $\phi(Z_i)$ with respect to $Z_0$, as described in Section 3. Therefore, the entire embedding for each graph $G_i, i \in \{1, \ldots, M\}$, is obtained by composing $\phi(\cdot)$ and $h(\cdot)$, i.e., $\psi(G_i) = \phi(h(G_i))$. Figure 2 visualizes this process.

### 4.1 NODE EMBEDDING

There are many choices for node embedding methods (Chami et al., 2020). These methods in general could be parametric or non-parametric, e.g., as in propagation/diffusion-based embeddings. Parametric embeddings are often implemented via a GNN encoder. The encoder can capture different graph properties depending on the type of supervision (e.g., supervised or unsupervised). Self-supervised embedding methods have also been recently shown to be promising (Hu* et al., 2020).

In this paper, for our node embedding process $h(\cdot)$, we follow a similar non-parametric propagation/diffusion-based encoder as in (Togninalli et al., 2019). One of the appealing advantages of this framework is its simplicity, as there are no trainable parameters involved. In short, given a graph $G = (\mathcal{V}, \mathcal{E})$ with node features $\{x_v\}_{v \in \mathcal{V}}$ and scalar edge features $\{w_{uv}\}_{(u,v) \in \mathcal{E}}$, we use the

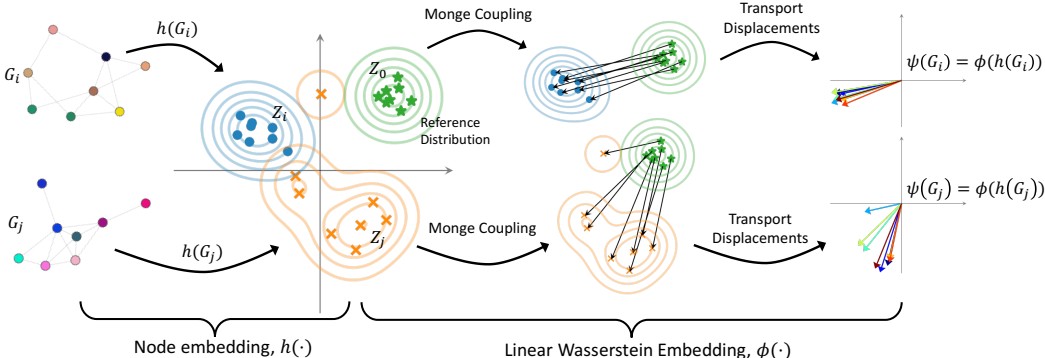

Figure 2: Our proposed graph embedding framework, WEGL, combines node embedding methods with the linear Wasserstein embedding framework described in Section 3. Given a graph $G_i = (\mathcal{V}_i, \mathcal{E}_i)$, we first embed the graph nodes into a $d$-dimensional Hilbert space and obtain an array of node embeddings, denoted by $h(G_i) = Z_i \in \mathbb{R}^{|V_i| \times d}$. We then calculate the linear Wasserstein embedding of $Z_i$ with respect to a reference $Z_0$, i.e., $\phi(Z_i)$, to derive the final graph embedding.

following instantiation of equation 12 to define the combining function as

$$x_v^{(l)} = \sum_{u \in \mathcal{N}_v \cup \{v\}} \frac{w_{uv}}{\sqrt{\deg(u)\deg(v)}} \cdot x_u^{(l-1)}, \ \forall l \in \{1, \ldots, L\}, \ \forall v \in \mathcal{V}, \quad (7)$$

where for any node $v \in \mathcal{V}$, its degree $\deg(v)$ is defined as its number of neighbors in $G$ augmented with self-connections, i.e., $\deg(v) \triangleq 1 + |\mathcal{N}_v|$. Note that the normalization of the messages between graph nodes by the (square root of) the two end-point degrees in equation 7 have also been used in other architectures, including GCN (Kipf & Welling, 2016). For the cases where the edge weights are not available, including self-connection weights $\{w_{vv}\}_{v \in \mathcal{V}}$, we set them to one. In Appendix A.5, we show how we use an extension of equation 7 to treat graphs with multiple edge features/labels. Finally, we let $z_v = g\left(\{x_v^{(l)}\}_{l=0}^L\right)$ represent the resultant embedding for each node $v \in \mathcal{V}$, where $g(\cdot)$ is a local pooling process on a single node (not a global pooling), e.g., concatenation or averaging.

### 4.2 CALCULATION OF THE REFERENCE DISTRIBUTION

To calculate the reference distribution, we use the $k$-means clustering algorithm on $\bigcup_{i=1}^M Z_i$ with $N = \left\lfloor \frac{1}{M} \sum_{i=1}^M N_i \right\rfloor$ centroids. Alternatively, one can calculate the Wasserstein barycenter (Cuturi & Doucet, 2014) of the node embeddings or simply use $N$ samples from a normal distribution. While approximation of the 2-Wasserstein distance in the tangent space depends on the reference distribution choice, surprisingly, we see a stable performance of WEGL for different references. In Appendix A.2, we compare the performance of WEGL with respect to different references.

## 5 EXPERIMENTAL EVALUATION

In this section, we discuss the evaluation results of our proposed algorithm on multiple benchmark graph classification datasets. We use the PyTorch Geometric framework (Fey & Lenssen, 2019) for implementing WEGL. In all experiments, we use scikit-learn for the implementation of our downstream classifiers on the embedded graphs (Buitinck et al., 2013).

### 5.1 MOLECULAR PROPERTY PREDICTION ON THE OPEN GRAPH BENCHMARK

We first evaluate our algorithm on the molecular property prediction task on the `ogbg-molhiv` dataset. This dataset is part of the Open Graph Benchmark (Hu et al., 2020), which involves node-level, link-level, and graph-level learning and prediction tasks on multiple datasets spanning diverse problem domains. The `ogbg-molhiv` dataset, in particular, is a molecular tree-like dataset, consisting of $41,127$ graphs, with an average number of $25.5$ nodes and $27.5$ edges per graph. Each graph is a molecule, with nodes representing atoms and edges representing bonds between them, and

| | Method | Validation ROC-AUC (%) | Test ROC-AUC (%) |
|---|---|---|---|
| GNN | GCN (Kipf & Welling, 2016) | 83.8 ± 0.9 | 76.0 ± 1.2 |
| | GIN + Virtual Node (Xu et al., 2019) | **84.8 ± 0.7** | 77.1 ± 1.5 |
| | DeeperGCN (Li et al., 2020) | 84.3 ± 0.6 | 78.6 ± 1.2 |
| | HIMP (Fey et al., 2020) | - | 78.8 ± 0.8 |
| | GCN + GraphNorm (Cai et al., 2020) | 79.0 ± 1.1 | 78.8 ± 1.0 |
| Ours | WEGL + Random Forest | 79.2 ± 2.2 | 75.5 ± 1.5 |
| | WEGL + Virtual Node + Random Forest | 81.9 ± 1.3 | 76.5 ± 1.8 |
| | WEGL + Virtual Node + AutoML | 81.6 ± 0.6 | **79.1 ± 0.3** |
| | GAP + Virtual Node + Random Forest | 74.9 ± 2.4 | 72.1 ± 1.7 |

Table 1: Graph classification results on the `ogbg-molhiv` dataset. The results for GCN and GIN are reported from (Hu et al., 2020). The best validation and test results are shown in **bold**.

it includes both node and edge attributes, characterizing the atom and bond features. The goal is to predict a binary label indicating whether or not a molecule inhibits HIV replication.

To train and evaluate our proposed method, we use the scaffold split provided by the dataset, and report the mean and standard deviation of the results across 10 different random seeds. We perform a grid search over a set of hyperparameters and report the configuration that leads to the best validation performance. We also report a *virtual node* variant of the graphs in our evaluations, where each graph is augmented with an additional node that is connected to all the original nodes in the graph. This node serves as a shortcut for message passing among the graph nodes, bringing any pair of nodes within at most two hops of each other. The complete implementation details can be found in Appendix A.5.

Table 1 shows the evaluation results of WEGL on the `ogbg-molhiv` dataset in terms of the ROC-AUC (i.e., Receiver Operating Characteristic Area Under the Curve), alongside multiple GNN-based baseline algorithms. Specifically, we show the results using two classifiers: A random forest classifier (Breiman, 2001) and an automated machine learning (AutoML) classifier using the Auto-Sklearn 2.0 library (Feurer et al., 2020). As the table demonstrates, while WEGL embeddings combined with random forest achieve a decent performance level, using AutoML further enhances the performance and achieves state-of-the-art test results on this dataset, showing the high expressive power of WEGL in large-scale graph datasets, without the need for end-to-end training.

Moreover, as an ablation study, we report the evaluation results on the `ogbg-molhiv` dataset using a random forest classifier, where the Wasserstein embedding module after the node embedding process is replaced with global average pooling (GAP) among the output node embeddings of each graph to derive the graph-level embedding. As the table demonstrates, there is a significant performance drop when using GAP graph embedding, which indicates the benefit of our proposed graph embedding method as opposed to average readout.

## 5.2 TUD Benchmark Datasets

We also consider a set of social network, bioinformatics and molecule graph datasets (Kersting et al., 2020). The social network datasets (`IMDB-BINARY`, `IMDB-MULTI`, `COLLAB`, `REDDIT-BINARY`, and `REDDIT-MULTI-5K`) lack both node and edge features. Therefore, in these datasets we use a one-hot representation of the node degrees as their initial feature vectors, as also used in prior work, e.g., (Xu et al., 2019). To handle the large scale of the `REDDIT-BINARY` and `REDDIT-MULTI-5K` and datasets, we clip the node degrees at 500.

Moreover, for the molecule (`PTC-MR`) and bioinformatics (`ENZYMES` and `PROTEINS`) datasets, we use the readily-provided node labels in (Kersting et al., 2020) as the initial node feature vectors. Besides, for `PTC-MR` which has edge labels, as explained in Appendix A.5, we use an extension of equation 7 to use the one-hot encoded edge features in the diffusion process. To evaluate the performance of WEGL, we follow the methodology used in (Yanardag & Vishwanathan, 2015; Niepert et al., 2016; Xu et al., 2019), where for each dataset, we perform 10-fold cross-validation with random splitting on the entire dataset, conducting a grid search over the desired set of hyperparameters as mentioned in Appendix A.5, and we then report the mean and standard deviation of the validation accuracies achieved during cross-validation. For this experiment, we use two ensemble classifiers,

| | Method | IMDB-B | IMDB-M | COLLAB | RE-B | RE-M5K | PTC-MR | ENZYMES | PROTEINS |
|---|---|---|---|---|---|---|---|---|---|
| GNN | DGCNN (Zhang et al., 2018a) | 69.2±3.0 | 45.6±3.4 | 71.2±1.9 | 87.8±2.5 | 49.2±1.2 | 58.6 | 38.9±5.7 | 72.9±3.5 |
| | GraphSAGE (Hamilton et al., 2017) | 68.8±4.5 | 47.6±3.5 | 73.9±1.7 | 84.3±1.9 | 50.0±1.3 | 63.9±7.7 | - | 75.9±3.2 |
| | GIN (Xu et al., 2019) | 75.1±5.1 | 2.3±2.8 | 80.2±1.9 | **92.4±2.5** | **57.5±1.5** | 64.6±7.0 | - | 76.2±2.8 |
| | GNTK (Du et al., 2019) | **76.9±3.6** | **52.8±4.6** | **83.6±1.0** | - | - | **67.9±6.9** | - | 75.6±4.2 |
| | CapsGNN (Xinyi & Chen, 2019) | 73.1±4.8 | 50.3±2.6 | 79.6±0.9 | - | 52.9±1.5 | - | 54.7±5.7 | **76.3±3.6** |
| | GraphNorm (Cai et al., 2020) | **76.0±3.7** | - | 80.2±1.0 | 93.5±2.1 | - | 64.9±7.5 | - | **77.4±4.9** |
| GK | DGK (Yanardag & Vishwanathan, 2015) | 67.0±0.6 | 44.6±0.5 | 73.1±0.3 | 78.0±0.4 | 41.3±0.2 | 57.3±1.1 | 27.1±0.8 | 71.7±0.5 |
| | WL (Shervashidze et al., 2011) | 73.8±3.9 | 49.8±0.5 | 74.8±0.2 | 68.2±0.2 | 51.2±0.3 | 57.0±2.0 | 53.2±1.1 | 72.9±0.6 |
| | RetGK (Zhang et al., 2018b) | 71.0±0.6 | 46.7±0.6 | 73.6±0.3 | 90.8±0.2 | 54.2±0.3 | **67.9±1.4** | 59.1±1.1 | 75.2±0.3 |
| | AWE (Ivanov & Burnaev, 2018) | 74.5±5.8 | 51.5±3.6 | 73.9±1.9 | 87.9±2.5 | 50.5±1.9 | - | 35.8±5.9 | - |
| | WWL (Togninalli et al., 2019) | 74.4±0.8 | - | - | - | - | 66.3±1.2 | **59.1±0.8** | 74.3±0.6 |
| Ours | WEGL + SVM-RBF | 73.4±2.5 | 51.7±3.1 | 78.6±1.0 | 92.1±1.9 | **56.1±2.3** | 63.4±5.3 | 57.3±4.2 | 76.0±4.4 |
| | WEGL + Random Forest | **75.4±5.0** | **52.0±4.1** | 79.8±1.5 | 92.0±0.8 | 55.1±2.5 | **67.5±7.7** | **60.5±5.9** | **76.5±4.2** |
| | WEGL + GBDT | 75.2±5.0 | **52.3±2.9** | **80.6±2.0** | **92.9±1.9** | 55.4±1.6 | 66.2±6.9 | **60.0±6.3** | 76.3±3.9 |

Table 2: Graph classification accuracy (%) of our method and comparison with the state-of-the-art GNNs and graph kernels (GKs) on various TUD graph classification tasks. The results for DGCNN are reported from (Errica et al., 2020). The top-three performers on each dataset are shown in **bold**.

namely Random Forest and Gradient Boosted Decision Tree (GBDT), together with kernel-SVM with an RBF kernel (SVM-RBF). Given that the Euclidean distance in the embedding space approximates the 2-Wasserstein distance, the SVM-RBF classification results are comparable with those reported by Togninalli et al. (2019).

Table 2 shows the classification accuracies achieved by WEGL on the aforementioned datasets as compared with several GNN and graph kernel baselines, whose results are extracted from the corresponding original papers. As the table demonstrates, our proposed algorithm achieves either state-of-the-art or competitive results across all the datasets, and in particular, it is among the top-three performers in all of them. This shows the effectiveness of the proposed linear Wasserstein embedding for learning graph-level properties across different domains.

## 5.3 COMPUTATION TIME

As mentioned before, one of the most important advantages of WEGL as compared to other graph representation learning methods is its algorithmic efficiency. To evaluate that, we compare the wall-clock training and inference times of WEGL with those of GIN and the Wasserstein Weisfeiler-Lehman (WWL) graph kernel on five different TUD datasets (IMDB-B, MUTAG, PTC-MR, PROTEINS, and NCI1)[2]. For WEGL and WWL, we use the exact linear programming solver (as opposed to the entropy-regularized version). We carry out our experiments for WEGL and WWL on a 2.3 GHz Intel® Xeon® E5-2670 v3 CPU, while we use a 16 GB NVIDIA® Tesla® P100 GPU for GIN. As a reference, we implement GIN on the aforementioned CPU hardware as well.

Figure 3 shows how the training and inference run-times of WEGL, WWL and GIN scale with the number of graphs in the dataset, the average number of nodes per graph, and the average number of edges per graph. As the figure illustrates, while having similar or even better performance, training WEGL is several orders of magnitude faster than WWL and GIN, especially for datasets with larger numbers of graphs. Note that training WWL becomes very inefficient as the number of graphs in the dataset increases due to the pairwise distance calculation between all the graphs.

During inference, WEGL is slightly slower than GIN, when implemented on a GPU. However, on most datasets, WEGL is considerably faster in inference as compared to GIN implemented on a CPU. Moreover, both algorithms are significantly faster than WWL. Using GPU-accelerated implementations of the diffusion process in equation 7 and the entropy-regularized transport problem could potentially further enhance the computational efficiency of WEGL during inference.

---

[2]The complete classification results of WEGL on these datasets can be found in Appendix A.6.

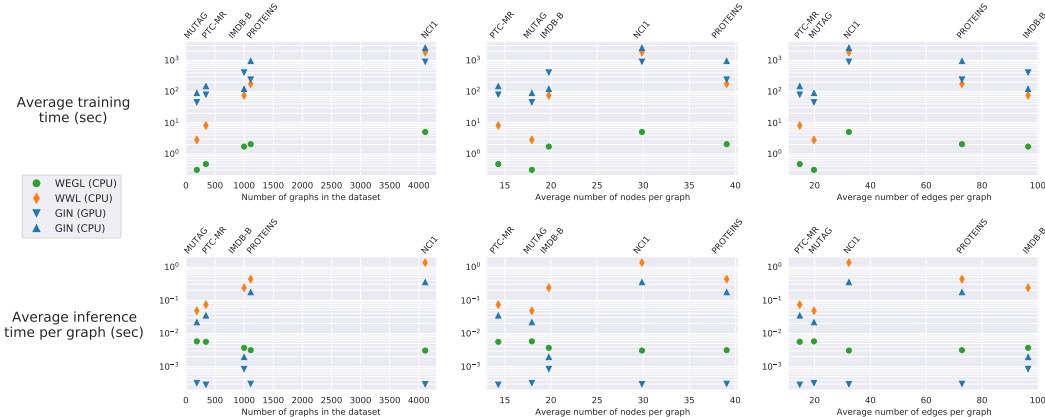

Figure 3: Average wall-clock time comparison of our proposed method, WEGL with WWL (Togninalli et al., 2019) and GIN (Xu et al., 2019). WEGL and WWL were implemented on a 2.3 GHz Intel® Xeon® E5-2670 v3 CPU. Moreover, GIN was implemented twice, once using the aforementioned CPU and once using a 16 GB NVIDIA® Tesla® P100 GPU, with 5 hidden layers, 300 training epochs and a batch size of 16 in both cases.

## 6 CONCLUSION

We considered the problem of graph property prediction and introduced the linear Wasserstein Embedding for Graph Learning, which we denoted as WEGL. Similar to (Togninalli et al., 2019), our approach also relies on measuring the Wasserstein distances between the node embeddings of graphs. Unlike (Togninalli et al., 2019), however, we further embed the node embeddings of graphs into a Hilbert space, in which their Euclidean distance approximates their 2-Wasserstein distance. WEGL provides two significant benefits: i) it has linear complexity in the number of graphs (as opposed to the quadratic complexity of (Togninalli et al., 2019)), and ii) it enables the application of any ML algorithm of choice, such as random forest, gradient boosted decision tree, or even AutoML. We demonstrated WEGL's superior performance and highly efficient training on a wide range of benchmark datasets, including the `ogbg-molhiv` dataset and the TUD graph classification tasks.

## ACKNOWLEDGEMENT

We gratefully acknowledge funding by the United States Air Force under Contract No. FA8750-19-C-0098. Gustavo K. Rohde also acknowledges funding by NIH grant GM130825. Any opinions, findings, and conclusions or recommendations expressed in this material are those of the author(s) and do not necessarily reflect the views of the United States Air Force and DARPA.

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

# A APPENDIX

Here we provide further details on the theoretical aspect of WEGL, our implementation details, and the sensitivity of the results to the choice of reference distribution. We also share our implementation code on the `ogbg-molhiv` dataset to help with the review process.

## A.1 DETAILED DISCUSSION ON LINEAR WASSERSTEIN EMBEDDING

The linear Wasserstein embedding used in WEGL is based on the Linear Optimal Transport (LOT) framework introduced in (Wang et al., 2013). The main idea is to compute the "projection" of the manifold of probability measures to the tangent space at a fixed reference measure. In particular, the tangent space at measure $\mu_0$ is the set of vector fields $\mathcal{V}_{\mu_0} = \{v : \mathcal{Z} \to \mathbb{R}^d \mid \int_{\mathcal{Z}} |v(z)|^2 d\mu_0(z) < \infty\}$ such that the inner product is the weighted $\ell_2$:

$$\langle v_i, v_j \rangle_{\mu_0} = \int_{\mathcal{Z}} v_i(z) \cdot v_j(z) d\mu_0(z). \tag{8}$$

We can then define $v_i(\cdot) \triangleq f(\cdot) - id(\cdot)$, where $f(\cdot)$ is the optimal transport map from $\mu_0$ to $\mu_i$. Note that $v_i \in \mathcal{V}_{\mu_0}$, $v_0 = 0$, and

$$\|v_i - v_0\|_{\mu_0}^2 = \|v_i\|_{\mu_0}^2 = \langle v_i, v_i \rangle_{\mu_0} = \int_{\mathcal{Z}} \|f(z) - z\|^2 d\mu_0(z) = \mathcal{W}_2(\mu_i, \mu_0). \tag{9}$$

In the paper, we use $\phi(\mu_i) = v_i \sqrt{p_0}$ to turn the weighted-$\ell_2$ into $\ell_2$.

The discussion above assumes an absolutely continuous reference measure $\mu_0$. A more interesting treatment of the problem is via the *generalized geodesics* defined in (Ambrosio et al., 2008), connecting $\mu_i$ and $\mu_j$ and enabling us to use discrete reference measures. Following the notation in (Ambrosio et al., 2008), given the reference measure $\mu_0$, let $\Gamma(\mu_i, \mu_0)$ be the set of transport plans between $\mu_i$ and $\mu_0$, and let $\Gamma(\mu_i, \mu_j, \mu_0)$ be the set of all measures on the product space $\mathcal{Z} \times \mathcal{Z} \times \mathcal{Z}$ such that the marginals over $\mu_i$ and $\mu_j$ are $\Gamma(\mu_j, \mu_0)$ and $\Gamma(\mu_i, \mu_0)$, respectively. Then the linearized optimal transport distance is defined as

$$d_{\text{LOT},\mu_0}^2(\mu_i, \mu_j) = \inf_{\gamma \in \Gamma(\mu_i, \mu_j, \mu_0)} \int_{\mathcal{Z} \times \mathcal{Z} \times \mathcal{Z}} \|z - z'\|^2 d\gamma(z, z', z''). \tag{10}$$

In a discrete setting, where $\mu_i = \frac{1}{N_i} \sum_{n=1}^{N_i} \delta_{z_n}$, $\mu_j = \frac{1}{N_j} \sum_{m=1}^{N_j} \delta_{z'_m}$, and $\mu_0 = \frac{1}{N} \sum_{l=1}^{N} \delta_{z''_l}$, we have

$$d_{\text{LOT},\mu_0}^2(\mu_i, \mu_j) = \min_{\gamma \in \Gamma(\mu_i, \mu_j, \mu_0)} \frac{1}{N_i N_j N} \sum_{n=1}^{N_i} \sum_{m=1}^{N_j} \sum_{l=1}^{N} \gamma_{nml} \|z_n - z'_m\|^2. \tag{11}$$

See Figure 4a for a depiction of Equation equation 11's meaning. Finally, the idea of barycenteric projection used to approximate Monge couplings and provide a fixed-size representation is shown in Figure 4b.

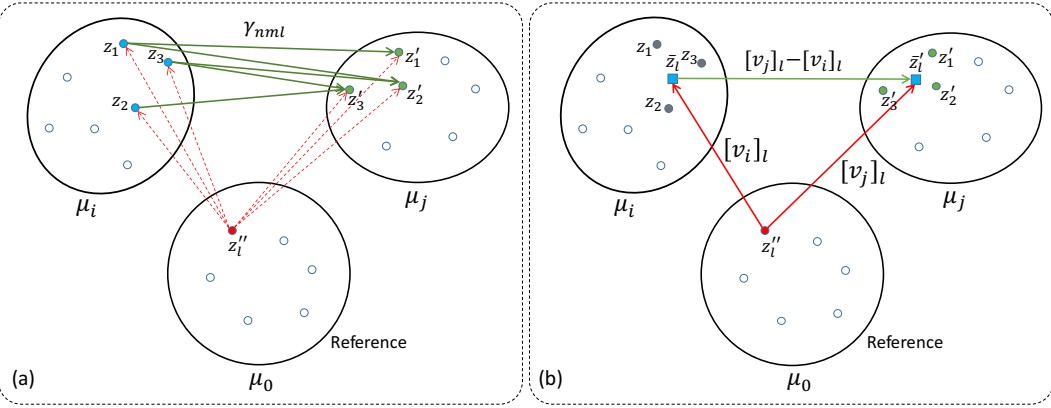

Figure 4: Illustration of (a) the meaning behind $\gamma_{nml}$ used in the LOT distance in equation 11, and (b) the idea of the barycenteric projection, which provides a fixed-size representation (i.e., of size $N$).

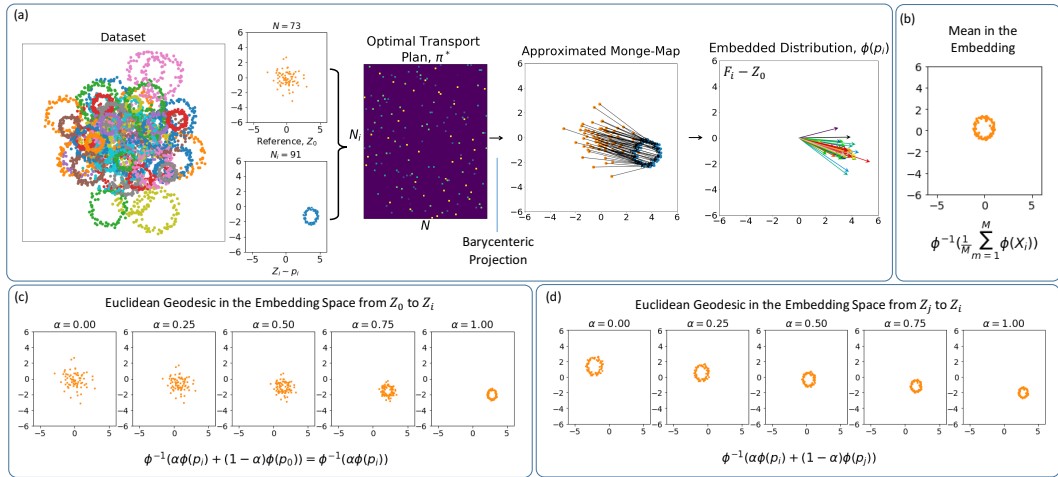

Figure 5: An experiment demonstrating the capability of the linear Wasserstein embedding. (a) A simple dataset consisting of shifted and scaled noisy ring distributions $\{p_i\}_{i=1}^M$, where we only observe samples $Z_i = [z_k^i \sim p_i]_{i=1}^{N_i}$ from each distribution, together with the process of obtaining the linear Wasserstein embedding with respect to a reference distribution. In short, for each distribution $p_i$, the embedding approximates the Monge-map (i.e., a vector field) from the reference samples $Z_0$ to the target samples $Z_i$ by a barycentric projection of the optimal transport plan. Adding samples in the embedding space corresponds to adding their vector fields, which can be used to calculate (b) the mean distribution in the embedding space, i.e., $\phi^{-1}(\frac{1}{M}\sum_{i=1}^M \phi(p_i))$ and (c)-(d) the Euclidean geodesics in this space, i.e., $\phi^{-1}(\alpha\phi(p_i) + (1-\alpha)\phi(p_j))$ for $\alpha \in [0, 1]$. As can be seen, the calculated mean is the Wasserstein barycenter of the dataset, and the Euclidean geodesics in the embedding space follow the Wasserstein geodesics in the original space.

Next, to demonstrate the capability of the linear Wasserstein embedding, we present the following experiment. Consider a set of distributions $\{p_i\}_{i=1}^M$, where each $p_i$ is a translated and dilated ring distribution in $\mathbb{R}^2$, and $N_i$ samples are observed from $p_i$, where $N_i$ and $N_j$ could be different for $i \neq j$. We then consider a normal distribution as the reference distribution and calculate the linear Wasserstein embedding with respect to the reference (See Figure 5a). Given the pseudo-invertible nature of the embedding, to demonstrate the modeling capability of the framework, we calculate the mean in the embedding space (i.e., on the vector fields), and invert it to obtain the mean distribution $\bar{p}$. Figure 5b shows the calculated mean, indicating that the linear Wasserstein embedding framework has successfully retrieved a ring distribution as the mean. Finally, we calculate Euclidean geodesics in the embedding space (i.e., the convex combination of the vector fields) between $p_i$ and $p_0$, as well as between $p_i$ and $p_j$, and show the inverted geodesics in Figures 5c and 5d, respectively. As the figures demonstrate, the calculated geodesics follow the Wasserstein geodesics.

APPROXIMATION ERROR OF THE EMBEDDING

Given the Euclidean distance in the embedding space is a transport-based distance (i.e., the so-called LOT distance) that approximates the Wasserstein distance, a natural question arises about the approximation error. Here we point the reader to the recent work of Moosmüller & Cloninger (2020) in which the authors provide bounds on how well the Euclidean distance in the embedding space approximates the 2-Wasserstein distance. In particular, the authors show that:

$$\mathcal{W}_2(\mu_i, \mu_j) \leq \|\phi(\mu_i) - \phi(\mu_j)\|_2 \leq \mathcal{W}_2(\mu_i, \mu_j) + \|f_{\mu_i}^{\mu_j} - f_{\mu_0}^{\mu_j} \circ f_{\mu_i}^{\mu_0}\|_{\mu_i},$$

where $f_{\mu_i}^{\mu_j}$ is the optimal transport map from $\mu_i$ to $\mu_j$. This inequality simply indicates that the approximation error is caused by conditioning the transport map to be obtained by composition of the optimal transport maps from $\mu_i$ to $\mu_0$, and then from $\mu_0$ to $\mu_j$. More importantly, it can be shown that if $\mu_i$ and $\mu_j$ are shifted and scaled versions of the reference measure, $\mu_0$, then the embedding is isometric (See Figure 1 and 2 in Moosmüller & Cloninger (2020)).

Maybe a less interesting upper bound can also be obtained by the triangle inequality:

$$\mathcal{W}_2(\mu_i, \mu_j) \leq \|\phi(\mu_i) - \phi(\mu_j)\|_2 \leq \mathcal{W}_2(\mu_i, \sigma) + \mathcal{W}_2(\sigma, \mu_j),$$

which ensures some regularity of the embedding.

REGULARITY OF THE EMBEDDING

A good question was raised during the feedback period, about the regularity of the proposed embedding. The regularity of the graph embedding will depend on both the regularity of node-embedding and the Wasserstein embedding. In the following, we avoid the discussion on regularity of the node-embedding (as it is not the main focus of our work), and focus on the regularity of the Wasserstein embedding. To that end, we first point out several regularity characteristics pointed out in Appendix A of Moosmüller & Cloninger (2020). Most notably Theorem 4.2 in their paper, shows an *almost isometric* property when the distortions are within an $\epsilon$-tube around the set of shifts and scalings. In short, let $\mu \in P_2(\mathcal{Z})$, $R > 0$, $\epsilon > 0$, $\mathcal{E}$ be the set of all shifts and scalings, and

$$\mathcal{E}_{\mu,R} = \{h \in \mathcal{E} : \|h\|_\mu \leq R\},$$

and let

$$\mathcal{G}_{\mu,R,\epsilon} = \{g \in L^2(\mathcal{Z}, \mu) : \exists h \in \mathcal{E}_{\mu,R} : \|g - h\|_\mu \leq \epsilon\},$$

which is the $\epsilon$-tube around set of shifts and scalings. Now, assume $\mu_0 \in P_2(\mathcal{Z})$ is the reference measure and both $\mu$ and $\mu_0$ satisfy Caffarelli's regularity theorem. Then for $g_1, g_2 \in \mathcal{G}_{\mu,R,\epsilon}$ we have

$$0 \leq \|\phi(g_{1\#}\mu) - \phi(g_{2\#}\mu)\|_2 - \mathcal{W}_2(g_{1\#}\mu, g_{2\#}\mu) \leq C_{\mu_0,\mu,R}\epsilon + \bar{C}_{\mu_0,\mu,R}\epsilon^2,$$

where $C_{\mu_0,\mu,R}$ and $\bar{C}_{\mu_0,\mu,R}$ are constants depending on $\mu_0$, $\mu$, and $R$.

The results shown above can be used to derive regularity results for the linear Wasserstein embedding with respect to additive noise. We know that the addition of two random variables leads to a new random variable with its PDF being the convolution of the original PDFs. Therefore, for features $Z_i = [z_1, ..., z_{N_i}]^T$, let $\hat{Z}_i = [\underbrace{z_1 + e_1}_{\hat{z}_1}, ..., \underbrace{z_{N_i} + e_{N_i}}_{\hat{z}_{N_i}}]^T$ denote the noisy samples for $e_i \sim \eta$, where $\eta$ is the noise distribution. Then the noisy samples, $\hat{z}_i$, will be distributed according to $\hat{\mu}_i = \mu_i * \eta$. For instance, for the Gaussian additive noise, $\hat{\mu}_i$ is the smoothed version of $\mu_i$. Therefore, there exists a transport map in $g \in \mathcal{G}_{\mu_i,R,\epsilon}$ for which, $\hat{\mu}_i = g_{\#}\mu_i$ and $\|g - id\|_{\mu_i} = \mathcal{W}_2(\hat{\mu}_i, \mu_i) \leq \epsilon$, and we have:

$$0 \leq \|\phi(\mu_i) - \phi(\hat{\mu}_i)\|_2 \leq (C_{\mu_0,\mu_i,R} + 1)\epsilon + \bar{C}_{\mu_0,\mu_i,R}\epsilon^2.$$

## A.2 SENSITIVITY TO THE CHOICE OF REFERENCE DISTRIBUTION

To measure the dependency of WEGL on the reference distribution choice, we changed the reference to a normal distribution (i.e., data-independent). We compared the results of WEGL using the new reference distribution to that using a reference distribution calculated via $k$-means on the training set. We used the `ogbg-molhiv` dataset with initial node embedding of size 300 and 4 diffusion layers. We ran the experiment with 100 different random seeds, and measured the test ROC-AUC of WEGL calculated with the two aforementioned reference distributions. Figure 6 shows the results of this experiment, indicating that the choice of reference distribution is statistically insignificant.

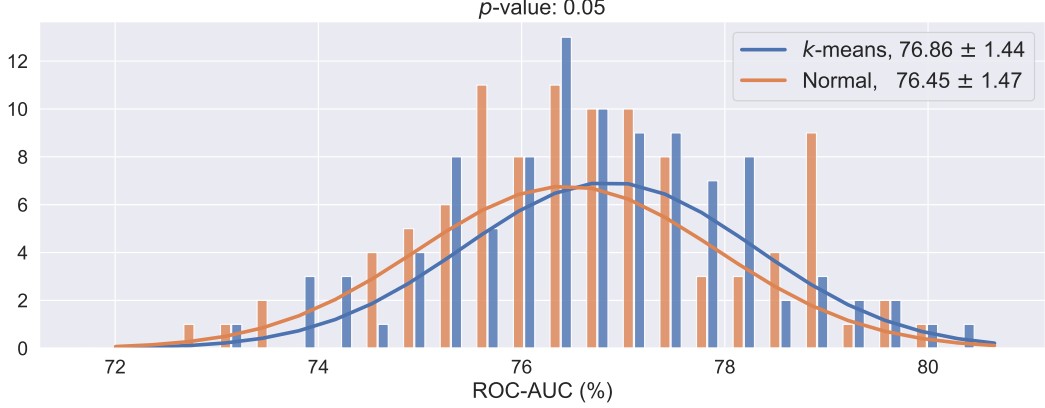

Figure 6: ROC-AUC (%) results on `ogbg-molhiv` dataset, when the reference distribution is calculated by $k$-means (Section 4.2) on the training dataset (denoted as $k$-means), compared to when it is fixed to be a normal distribution (denoted as Normal). With a $p$-value= 0.05, the choice of the template is statistically insignificant.

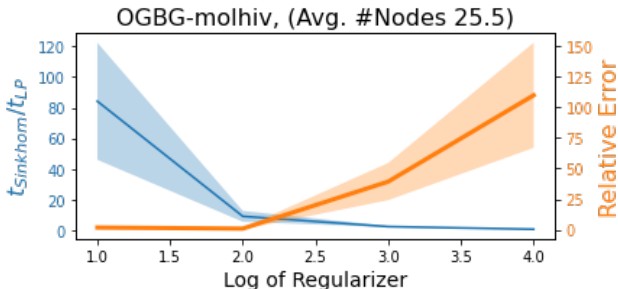

Figure 7: Comparing the performance of the linear programming (LP) solver with the Sinkhorn algorithm (on entropy regularized OT problem) on the `ogbg-molhiv` dataset for various regularization parameters.

### A.3 Linear Programming vs. Entropy Regularization

In this paper, we used the Python Optimal Transport (Flamary & Courty, 2017) for the calculation of the optimal transport plans. During the feedback period, a point came up regarding the entropy-regularized version of the OT problem (Cuturi & Doucet, 2014), which reduces the complexity of the linear programming problem from being cubic, in the number of nodes, to being quadratic, using the Sinkhorn algorithm. Given the Sinkhorn algorithm's iterative nature, the computational gain of the method is prominent when calculating the transportation problem between graphs with a large number of nodes (e.g., larger than $10^3$). However, the graph datasets used in this paper often have a small number of nodes (e.g., $< 50$). In these settings, linear programming is efficient. To obtain any computational gain using the Sinkhorn algorithm, one would need to use a large regularization coefficient, which reduces the Sinkhorn algorithm's precision.

Here we ran an experiment on the `ogbg-molhiv` dataset. We obtain the transport plans between the node embeddings and the reference distribution using the linear programming solver (using `ot.emd2` from (Flamary & Courty, 2017)) and the Sinkhorn algorithm for the entropy-regularized problem (using `ot.sinkhorn2` from (Flamary & Courty, 2017)). We measure the calculation time as well as the calculated distances for both algorithms. For the Sinkhorn algorithm, we used four different regularization values. We report the mean and standard deviation of calculation time ratio, i.e., $\frac{t_{Sinkhorn}}{t_{LP}}$ and the relative error of calculating the 2-Wasserstein distance, i.e., $\left| \frac{\mathcal{W}_{2,Sinkhorn} - \mathcal{W}_{2,LP}}{\mathcal{W}_{2,LP}} \right|$ in Figure 7. As the figure shows, due to the small graph size, the LP solver is efficient and little to no gain can be obtained using the Sinkhorn algorithm. Nevertheless, in the case of dealing with larger graph sizes, the entropy regularized formulation should be the definite choice. Finally, for the entropy-regularized OT problem, more efficient solvers have been proposed that outperform the Sinkhorn algorithm (Dvurechensky et al., 2018). However, given the acceptable performance of the linear programming solver (at least for the graph datasets in this paper), we did not find it necessary to seek more efficient solvers.

### A.4 Inner Working of GNNs

In its most general form, a GNN consists of $L$ hidden layers, where at the $l^{\text{th}}$ layer, each node $v \in \mathcal{V}$ aggregates and combines *messages* from its 1-hop neighboring nodes $\mathcal{N}_v$, resulting in the feature vector

$$x_v^{(l)} = \Psi_{\text{combine}} \left( x_v^{(l-1)}, \left\{ (x_u^{(l-1)}, w_{uv}) \right\}_{u \in \mathcal{N}_v} \right), \ \forall l \in \{1, \ldots, L\}, \ \forall v \in \mathcal{V}, \quad (12)$$

where $\Psi_{\text{combine}}(\cdot)$ denotes a parametrized and differentiable combining function.

At the input layer, each node $v \in \mathcal{V}$ starts with its initial feature vector $x_v^0 = x_v \in \mathbb{R}^F$, and the sequential application of GNN layers, as in equation 12, computes intermediate feature vectors $\{x_v^{(l)}\}_{l=1}^L$. At the GNN output, the feature vectors of all nodes from all layers go through a global pooling (i.e., readout) function $\Psi_{\text{readout}}(\cdot)$, resulting in the final graph embedding

$$\psi(G) = \Psi_{\text{readout}} \left( \left\{ x_v^{(l)} \right\}_{v \in \mathcal{V}, l \in \{0, \ldots, L\}} \right). \quad (13)$$

### A.5 IMPLEMENTATION DETAILS

To derive the node embeddings, we use the diffusion process in equation 7 for the datasets without edge features/labels, i.e., all the social network datasets (`IMDB-BINARY`, `IMDB-MULTI`, `COLLAB`, `REDDIT-BINARY`, `REDDIT-MULTI-5K`, and `REDDIT-MULTI-12K`) and four of the molecule and bioinformatics datasets (`ENZYMES`, `PROTEINS`, `D&D`, and `NCI1`). We specifically set $w_{uv} = 1$ for any $(u, v) \in \mathcal{E}$ and also for all self-connections, i.e., $w_{vv} = 1$, $\forall v \in \mathcal{V}$.

The remaining datasets contain edge labels that cannot be directly used with equation 7. Specifically, each edge in the `MUTAG` and `PTC-MR` datasets has a categorical label, encoded as a one-hot vector of dimension four. Moreover, in the `ogbg-molhiv` dataset, each edge has three categorical features indicating bond type (five categories), bond stereochemistry (six categories) and whether the bond is conjugated (two categories). We first convert each categorical feature to its one-hot representation, and then concatenate them together, resulting in a binary 13-dimensional feature vector for each edge.

In each of the three aforementioned datasets, for any edge $(u, v) \in \mathcal{E}$, let us denote its binary feature vector by $w_{uv} \in \{0, 1\}^E$, where $E$ is equal to 4, 4, and 13 for `MUTAG`, `PTC-MR`, and `ogbg-molhiv`, respectively. We then use the following extension of the diffusion process in equation 7,

$$x_v^{(l)} = \sum_{u \in \mathcal{V}} \left( \sum_{e=1}^{E} \frac{w_{uv,e}}{\sqrt{\deg_e(u)\deg_e(v)}} \right) x_u^{(l-1)}, \ \forall l \in \{1, \ldots, L\}, \ \forall v \in \mathcal{V}, \tag{14}$$

where for any $e \in \{1, \ldots, E\}$, $w_{uv,e}$ denotes the $e^{\text{th}}$ element of $w_{uv}$, and for any node $v \in \mathcal{V}$, we define $\deg_e(v)$ as its degree over the $e^{\text{th}}$ elements of the edge features; i.e., $\deg_e(v) \triangleq \sum_{u \in \mathcal{V}} w_{uv,e}$. We assign vectors of all-one features to the self-connections in the graph; i.e., $w_{vv,e} = 1$, $\forall v \in \mathcal{V}, \forall e \in \{1, \ldots, E\}$. Note that the formulation of the diffusion process in equation 14 can be seen as an extension of equation 7, where the underlying graph with multi-dimensional edge features is broken into $E$ *parallel* graphs with non-negative single-dimensional edge features, and the parallel graphs perform message passing at each round/layer of the diffusion process.

For the `ogbg-molhiv` experiments in which virtual nodes were appended to the original molecule graphs, we set the initial feature vectors of all virtual nodes to all-zero vectors. Moreover, for any graph $G_i$ in the dataset with $|\mathcal{V}_i|$ nodes, we set the edge features for the edge between the virtual node $v_{\text{virtual}}$ and each of the original graph nodes $u \in \mathcal{V}_i$ as $w_{uv_{\text{virtual}},e} = \frac{1}{|\mathcal{V}_i|}, \forall e \in \{1, \ldots, E\}$. The normalization by the number of graph nodes is included so as to regulate the degree of the virtual node used in equation 14. We also include the resultant embedding of the virtual node at the end of the diffusion process in the calculation of the graph embedding $\psi(G_i)$.

In the experiments conducted on each dataset, once the node embeddings are derived from the diffusion process, we standardize them by subtracting the mean embedding and dividing by the standard deviation of the embeddings, where the statistics are calculated based on all the graphs in the dataset. Moreover, to reduce the computational complexity of estimating the graph embeddings for the `ogbg-molhiv` dataset, we further apply a 20-dimensional PCA on the node embeddings.

### HYPERPARAMETERS

We use the following set of hyperparameters to perform a grid search over in each of the experiments:

- **Random Forest**: `min_samples_leaf` $\in \{1, 2, 5\}$, `min_samples_split` $\in \{2, 5, 10\}$, and `n_estimators` $\in \{25, 50, 100, 150, 200\}$.
- **Gradient Boosted Decision Tree (GBDT)**: `min_samples_leaf` $\in \{1, 2, 5\}$, `min_samples_split` $\in \{2, 5, 10\}$, `n_estimators` $\in \{25, 50, 100, 150, 200\}$, and `max_depth` $\in \{1, 3, 5\}$.
- **SVM-Linear and SVM-RBF**: $C \in \{10^{-2}, \ldots, 10^5\}$.
- **Multi-Layer Perceptron (MLP)**:
  `hidden_layer_sizes` $\in \{(128), (256), (128, 64), (256, 128)\}$.
- **Auto-ML**: Auto-Sklearn 2.0 searches over a space of 42 hyperparameters using Bayesian optimization techniques, as mentioned in Feurer et al. (2020).
- **Number of Diffusion Layers in equation 7 and equation 14**: $L \in \{3, \ldots, 8\}$.

- **Initial Node Feature Dimensionality (for `ogbg-molhiv` only)**: $\{100, 300, 500\}$.
- **Node Embedding Type**: For a graph with $F$-dimensional initial node features, we consider using the following three types of node embedding:

  - Concat: $z_v = \left[ x_v^{(0)} \| x_v^{(1)} \| \ldots \| x_v^{(L)} \right] \in \mathbb{R}^{(L+1)F}$, where $\|$ denotes concatenation.
  - Average: $z_v = \frac{1}{L+1} \sum_{l=0}^{L} x_v^{(l)} \in \mathbb{R}^{F}$.
  - Final: $z_v = x_v^{(L)} \in \mathbb{R}^{F}$.

## A.6 TUD BENCHMARK — COMPLETE RESULTS

Here, we report the comprehensive set of classification results of our proposed method, WEGL, for each of the node embedding types mentioned in Section A.5, using five different classifiers: Linear SVM, Kernel-SVM (SVM-RBF), Gradient Boosted Decision Trees (GBDT), Multi-Layer Perceptron (MLP), and Random Forest (RF). The results are shown in Table 3.

| Classifier + Embedding Type | | IMDB-B | IMDB-M | COLLAB | RE-B | RE-M5K | PTC-MR | ENZYMES | PROTEINS | MUTAG | NCI |
|---|---|---|---|---|---|---|---|---|---|---|---|
| SVM - Linear | Concat | 69.1 ± 6.1 | 41.3 ± 1.8 | 72.7 ± 1.9 | 91.3 ± 1.8 | 55.6 ± 2.5 | 61.6 ± 5.8 | 50.0 ± 5.8 | 74.1 ± 2.3 | 83.6 ± 9.5 | 58.9 ± 7.2 |
| | Avg | 72.7 ± 5.5 | 51.7 ± 3.4 | 76.4 ± 1.3 | 79.8 ± 3.1 | - | 62.8 ± 4.9 | 39.3 ± 7.7 | 74.2 ± 3.5 | 82.0 ± 7.4 | 56.2 ± 3.6 |
| | Final | 72.5 ± 3.7 | 51.1 ± 6.3 | - | - | - | 62.2 ± 6.6 | 37.5 ± 5.5 | 73.1 ± 2.1 | 81.9 ± 5.4 | 57.4 ± 5.0 |
| SVM - RBF | Concat | 71.9 ± 2.4 | 49.2 ± 3.1 | 75.8 ± 2.6 | 92.1 ± 1.9 | 56.1 ± 2.3 | 62.5 ± 5.8 | 57.3 ± 4.2 | 76.0 ± 4.4 | 84.0 ± 7.6 | 67.8 ± 2.1 |
| | Avg | 73.4 ± 2.5 | 50.9 ± 2.7 | 78.6 ± 1.0 | 80.5 ± 2.6 | - | 63.4 ± 5.3 | 50.8 ± 5.0 | 75.7 ± 4.4 | 83.0 ± 8.4 | 65.1 ± 2.1 |
| | Final | 72.8 ± 3.8 | 51.7 ± 3.1 | - | - | - | 62.8 ± 8.2 | 52.7 ± 5.0 | 75.8 ± 4.7 | 82.4 ± 7.9 | 64.9 ± 2.3 |
| GBDT | Concat | 74.5 ± 4.2 | 51.9 ± 2.8 | 80.6 ± 2.0 | 92.9 ± 1.9 | 55.4 ± 1.6 | 63.4 ± 5.4 | 60.0 ± 6.3 | 76.3 ± 3.9 | 89.3 ± 6.6 | 78.4 ± 1.6 |
| | Avg | 75.2 ± 5.0 | 52.3 ± 2.9 | - | 88.8 ± 1.9 | - | 65.2 ± 6.0 | 58.5 ± 6.5 | 75.9 ± 2.7 | 87.8 ± 6.2 | 78.2 ± 2.9 |
| | Final | 75.1 ± 2.0 | 52.1 ± 4.0 | - | - | - | 66.2 ± 6.9 | 57.7 ± 5.6 | 75.9 ± 2.8 | 87.2 ± 10.0 | 76.4 ± 2.5 |
| MLP | Concat | 72.0 ± 3.5 | 48.9 ± 3.1 | 73.1 ± 2.4 | 86.9 ± 3.6 | 51.5 ± 1.6 | 60.2 ± 6.5 | 57.8 ± 5.4 | 72.5 ± 4.2 | 82.0 ± 6.2 | 61.3 ± 4.6 |
| | Avg | 72.0 ± 4.0 | 48.5 ± 3.7 | 78.6 ± 1.0 | 76.2 ± 2.3 | - | 61.1 ± 5.0 | 55.7 ± 5.6 | 74.1 ± 3.0 | 82.0 ± 8.5 | 60.0 ± 3.8 |
| | Final | 72.6 ± 3.5 | 50.3 ± 3.8 | - | - | - | 61.6 ± 6.3 | 55.5 ± 5.2 | 72.8 ± 4.1 | 81.9 ± 5.9 | 59.9 ± 3.7 |
| RF | Concat | 74.9±6.3 | 50.8±4.0 | 79.8±1.5 | 92.0±0.8 | 55.1±2.5 | 64.6±7.4 | 60.5±5.9 | 76.1±3.3 | 88.3±5.1 | 76.8±1.7 |
| | Avg | 74.4±5.6 | 52.0±4.1 | 78.8±1.4 | 87.8±3.0 | 53.3±2.0 | 67.5±7.7 | 58.7±6.9 | 75.8±3.6 | 86.2±5.8 | 76.6±1.1 |
| | Final | 75.4±5.0 | 51.7±4.6 | 79.1±1.1 | 87.5±2.0 | 53.2±1.8 | 66.3±6.5 | 57.5±5.5 | 76.5±4.2 | 86.2±9.5 | 75.9±1.2 |

WEGL

Table 3: Graph classification accuracy (%) of the Wasserstein embedding based on different classifiers and the three types of node embeddings, on various TUD graph classification tasks.

