# OpenReview forum: "Wasserstein Embedding for Graph Learning"
_ICLR.cc/2021/Conference — ICLR 2021 Poster_

### Official Review · AnonReviewer1 · 2020-10-25
**A smart, innovative embedding algorithm based on optimal transport**

**Rating:** 8
**Confidence:** 5

**Review:**

# Synopsis of the paper

This paper presents WEGL ("Wasserstein Embedding for Graph Learning"),
a technique for embedding graphs for graph classification. The primary
novelty of the paper lies in their smart choice of embedding, which
combines the expressivity of optimal transport paradigms (i.e. the
Wasserstein distance, in this case) with improved computational
performance; more precisely, the embeddings permit the use of effective
classification algorithms, as opposed to scaling quadratically with the
number of samples.

# Summary of the review

This is a well-written paper with a powerful algorithm and a strong
experimental section. It will make an excellent contribution to the
conference, and I am excited to endorse it!

There are some issues with the current version of the write-up, though,
which, if fixed, will make this an even stronger publication. My primary
concerns at present are:

1. The paper spends a lot of space in outlining background information
   that is not pertinent to the method. I appreciate the amount of
   details that are being packed into Section 2.1, but as a reader of
   this paper, I would prefer a more in-depth discussion of the method
   rather than a discussion of GNNs, which are used as mere comparison
   partners here. My suggestion would be to put some of these
   information into the appendix.

2. The section on Wasserstein distances could be streamlined. At
   present, too many different concepts are introduced; I feel that
   a non-expert reader will just be deterred, even though the remainder
   of the paper is very hands-on. Maybe some additional intuition could
   be provided here?

3. On the other hand, Section 3 is *missing* important information; in
   this section, I would be very much interested in knowing more about
   the theoretical properties of the method (and its empirical
   performance). The characteristics mentioned on p. 4 are intriguing,
   but it would improve the paper if a more detailed write-up would be
   provided; a reference with proofs for these properties would be
   equally helpful. I am fully aware that it is hard to satisfy both
   theory and experiments in a paper; I have some more detailed comments
   about what could be added (potentially in the supplementary materials,
   if the authors think that it detracts from the flow).

# Detailed Comments

- The point about the 'true metric' in the introduction is slightly
  ambiguous. My understanding is that one obtains a metric between the
  embedded feature vectors. This does *not* constitute a metric on the
  graphs, though, unless the embedding is injective. I think the
  sentence means to say that one obtains a metric in an embedding space
  and this metric serves as a proxy for the Wasserstein distance.

- In the related work section, I would suggest citing other variants of
  the WL algorithm that have recently emerged:

    - Morris et al.: *Weisfeiler and Leman Go Neural: Higher-Order Graph Neural Networks* (https://aaai.org/ojs/index.php/AAAI/article/view/4384)
    - Morris et al.: *Weisfeiler and Leman go sparse: Towards scalable higher-order graph embeddings* (https://arxiv.org/abs/1904.01543)
    - Rieck et al.: *A Persistent Weisfeiler-Lehman Procedure for Graph Classification* (http://proceedings.mlr.press/v97/rieck19a.html)

- The way a graph is defined in this paper could be misconstrued as
  *directed* upon first reading. Why not define edges in both directions
  by using a subset notation instead of a tuple notation?

- How are categorical attributes treated? In the supp. mat., the usual
  one-hot encoding is mentioned. This could be discussed earlier (i.e.
  on p. 2).

- When discussing the embedding, I would suggest briefly mentioning the
  concept of reproducing kernel Hilbert spaces (RKHS). Such a discussion
  will provide the relevant backdrop for this publication.

- As suggested above, I would shorten Section 2.1 somewhat (or put some
  of the text into the appendix). While it is good to provide some more
  details about the inner workings of GNNS, this is not required for the
  paper.

- Section 2.2 would benefit from more intuition; a lot of the
  terminology is introduced too tersely and not re-used. I fully
  appreciate that the paper is mathematically precise here, but at the
  same time, I would suggest a more streamlined introduction of the
  concepts that are necessary to define the embedding in the end.

- Section 3.1 and Figure 1 differ slightly in terms of their notation.
  I would suggest to harmonise the description here in order to be more
  consistent.

- For property 4 on p. 4, is it possible to quantify the strength of the
  approximation? How good is this approximation in practice, and what
  are the factors influencing it? I would suggest adding some more
  details here (or citations, if appropriate).

- To add to the previous point, I would in general like to know more
  about the stability properties of the full embedding. Can this easily
  be quantified? For example, what happens if I add some noise to the
  attributes? I would assume that stability is a function of the
  selected aggregation function *and* the Wasserstein embedding itself.
  Assuming that the former is fixed to be the function described by
  Togninalli et al., does the latter satisfy, for example, Lipschitz
  continuity? (I am asking this out of professional curiosity;
  understanding the inherent properties of the embedding seems key to me
  for us to understand better ways to generate such embeddings; the
  empirical performance of the proposed method speaks for itself, of
  course!)

- Section 3.2 is rather technical at present, making use of
  hitherto-undefined concepts. I would suggest improving this section
  by providing more intuition, relegating some of the more technical
  results to the appendix.

- Does the pseudo-invertibility of $\phi$ cause any problems in
  practice? It is my understanding that the embedding will not be
  injective in general anyway, or am I mistaken?

- How stable is the reference embedding? In Section 4.2, it is my
  understanding that the reference embedding is by default obtained from
  $k$-means, with $k$ being the average number of nodes. The appendix
  depicts the results for another data-independent reference embedding
  and states that there are no differences. Would this not suggest that
  the way the reference distribution is obtained is irrelevant? If so,
  why not use a normal distribution (which I would expect to be simpler
  to calculate than a $k$-means embedding) for all experiments?

  This point should be addressed somewhat better.

- Why is the virtual node inclusion necessary? Only for the
  simplification of the message passing itself? I am aware of this
  standard modification, but I do not see why the proposed algorithm
  could equally well work with the original graph.

- Why is the variance of the proposed method high for some of the data
  sets of the 'TUD' repository? Is this a consequence of the model that
  was picked for working with the embeddings?

# Style

This paper is well written; it was a pleasure to read and review. Here
are some minor suggestions to improve style/clarity:

- I would suggest so sort citations by some criterion (alphabetically,
  for example) when citing multiple authors.

- "See the recent survey" --> "see the recent survey"

- "such transport plan" --> "such a transport plan"

---

> ### Author Response · Authors · 2020-11-23
> **Response to Reviewer 1 (Part 1)**
>
> First, we appreciate the reviewer's positive evaluation of our work. Independent from the score, we would also like to genuinely thank the reviewer for the exemplar review, bringing up deep and insightful points, and providing constructive feedback to improve our paper. We have carefully considered the feedback, and below are our responses.
>
> ---
> **Q1.** Paper organization and missing references.
>
> **R.** Thank you for the helpful comment. As you mentioned, balancing the paper to be self-contained, engaging for a wide spectrum of audiences, and rigorous is challenging. We have restructured the paper to the extent possible to incorporate your points and added the mentioned references. We also reviewed the entire manuscript and, when citing multiple papers, ordered the citations chronologically, breaking the ties using the alphabetical order of the author names.
>
> ---
> **Q2.** The point about the 'true metric' in the introduction is slightly ambiguous.
>
> **R.** Thank you for pointing this out. Yes, we only have a true metric if the embedding is injective (both node embedding and the Wasserstein embedding), and in our case, neither of the embeddings are injective. We updated the sentence according to your suggestion.
>
> ---
> **Q3.** The way a graph is defined in this paper could be misconstrued as directed upon first reading. Why not define edges in both directions by using a subset notation instead of a tuple notation?
>
> **R.** To the best of our knowledge, a directed graph is a generalization of an undirected graph. Our formulation is general, and the diffusion process for feature propagation can work for directed graphs (as well as undirected graphs). Therefore, we kept the notation as it is, but added a footnote to Section 2.1 of the revised manuscript to highlight the applicability to both directed and undirected graphs.
>
> ---
> **Q4.** Leveraging the concept of reproducing kernel Hilbert spaces (RKHS) in the embedding's definition.
>
> **R.** We agree with the reviewer, and the connection to RKHS would make the paper more appealing to the researchers more familiar with kernel methods. We have updated the paper accordingly.
>
> ---
> **Q5.** For property 4 on p. 4, is it possible to quantify the strength of the approximation? How good is this approximation in practice, and what are the factors influencing it? I would suggest adding some more details here (or citations, if appropriate).
>
> **R.** This is a fantastic point! Indeed there are bounds on how well the Euclidean distance in the embedding space approximates the 2-Wasserstein distance. In particular, we mention the recent work of [Moosmüller and Collinger](https://arxiv.org/pdf/2008.09165.pdf), in which they show that:
>
> $\mathcal{W}_\{2\}(\mu_i,\mu_j)\leq \\| \phi(\mu_i)-\phi(\mu_j)\||_\{2\} \leq \mathcal{W}_\{2\}(\mu_i,\mu_j) + \\| f_\{\mu_i\}^\{\mu_j\} - f_\{\mu_0\}^\{\mu_j\} \circ f^{\mu_0\}_\{\mu_i\} \\|_\{\mu_i\},$
>
> where $f_{\mu_i}^{\mu_j}$ is the optimal transport map from $\mu_i$ to $\mu_j$. This inequality indicates that the approximation error is caused by conditioning the transport map to be obtained by composition of the optimal transport maps from $\mu_i$ to $\mu_0$, and then from $\mu_0$ to $\mu_j$. More importantly, it can be shown that if $\mu_i$ and $\mu_j$ are shifted and scaled versions of the reference measure, $\mu_0$, then the embedding is isometric (See Figure 1 and 2 in Moosmüller and Cloninger).
>
> Maybe a less interesting upper bound can also be obtained by the triangle inequality:
>
> $\mathcal{W}_2(\mu_i,\mu_j)\leq \\|\phi(\mu_i)-\phi(\mu_j)\\|_2\leq  \mathcal{W}_2(\mu_i,\sigma)+\mathcal{W}_2(\sigma,\mu_j),$
>
> which ensures some regularity of the embedding.
>
> ---

---

> ### Author Response · Authors · 2020-11-23
> **Response to Reviewer 1 (Part 2)**
>
> **Q6.** What regularity properties can be shown for the embedding?
>
> **R.** This is another good point, and we thank the reviewer for bringing it up. As you pointed out correctly, the regularity of the graph embedding will depend on the regularity of both the node embedding and the Wasserstein embedding. In the following, we avoid discussing the regularity of the node embedding process (as it is not the main focus of our work) and focus on the regularity of the Wasserstein embedding. To that end, we first point out several regularity characteristics pointed out in Appendix A of Moosmüller and Cloninger. Most notably, Theorem 4.2 in their paper shows an almost isometric property when the distortions are within an $\epsilon$-tube around the set of shifts and scalings. In short, let $\mu\in P_2(\mathcal{Z})$, $R>0$,  $\epsilon>0$, $\mathcal{E}$ be the set of all shifts and scalings, and
>
> $\mathcal{E}_\{\mu,R\}=\\{h\in\mathcal{E} : \|h\|_\mu\leq R\\},$
>
>  and let,
>
> $ \mathcal{G}_\{\mu,R,\epsilon\}=\\{g\in L^2(\mathcal{Z},\mu): \exists h\in \mathcal{E}_\{\mu,R\}:\\|g-h\\|_\mu \leq \epsilon\\},$
>
>  which is the $\epsilon$-tube around the set of shifts and scalings. Now, assume $\mu_0\in P_2(\mathcal{Z})$ is the reference measure and both $\mu$ and $\mu_0$ satisfy Caffarelli's regularity theorem. Then for $g_\{1\},g_\{2\} \in \mathcal{G}_\{ \mu,R,\epsilon \}$ we have,
>
> $ 0 \leq \\| \phi(\{g_\{1\}\}_\{\\#\} \mu)-\phi(\{g_\{2\}\}_\{\\#\} \mu)\\|_\{2\} -\mathcal{W}_\{2\}(\{g_\{1\}\}_\{\\#\} \mu,\{g_\{2\}\}_\{\\#\} \mu) \leq C_\{\mu_\{0\},\mu,R\}\epsilon +\bar\{C\}_\{\mu_\{0\},\mu,R\}\epsilon^\{2\}, $
>
> where $C_{\mu_0,\mu,R}$ and $\bar{C}_{\mu_0,\mu,R}$ are constants depending on $\mu_0$, $\mu$, and $R$.
>
> The results shown above can be used to derive regularity results for the linear Wasserstein embedding with respect to additive noise. We know that the addition of two random variables leads to a new random variable with its PDF being the convolution of the original PDFs. Therefore, for features $Z_i=[z_1,...,z_{N_i}]^T$, let $\hat{Z}_i=[\hat{z}_1,...,\hat{z}_\{N_i\}]^T=[z_1+e_1,..., z_\{N_i\}+e_\{N_i\}]^T$ denote the noisy samples for $e_i\sim \eta$, where $\eta$ is the noise distribution.
>
> Then the noisy samples, $\\hat{z}_i$ will be distributed according to $\\hat{\mu}_i=\mu_i*\eta$. For instance,  for the Gaussian additive noise, $\\hat{\mu}_i$ is the smoothed version of $\mu_i$. Therefore, there exists a transport map in $g\in\mathcal{G}_\{\mu_i, R,\epsilon\}$ for which,  $\\hat{\mu}_i=g_\{\\#\} \mu_i$  and $\\|g-id\\|_\{\mu_i\}=\mathcal{W}_2(\hat{\mu}_i,\mu_i)\leq \epsilon$, and we have:
>
> $0\leq \\|\phi(\mu_i)-\phi(\hat{\mu}_i)\\|_2 \leq (C_\{\mu_0,\mu_i,R\}+1)\epsilon +\bar{C}_\{\mu_0,\mu_i,R\}\epsilon^2.$
>
> We have added this discussion to the paper.
>
> ---
>
> **Q7.** Does the pseudo-invertibility cause any problems in practice?
>
> **R.** Correct. The pseudo-invertibility could become interesting for applications in which the node-embedding is also pseudo-invertible (e.g., the graph auto-encoders).
>
> ---
> **Q8.** How stable is the reference embedding? Why not using a normal distribution for all experiments?
>
> **R.** In light of our response to Q6, the reference distribution choice directly affects how well the Euclidean distance in the embedding space approximates the Wasserstein distance. In practice, we have seen that normal distribution as the reference consistently provides a lower accuracy than the k-means clustering scheme (which is consistent with Figure 6).
>
> ___
>
> **Q9.**  Why is the virtual node inclusion necessary? Only for the simplification of the message passing itself?
>
> **R.** As correctly pointed out by the reviewer, the addition of the virtual node simplifies message passing among nodes in datasets whose graphs resemble a tree-like structure, such as in the *ogbg-molhiv* dataset. According to the [official OGB leaderboard](https://ogb.stanford.edu/docs/graphprop/#ogbg-mol), the virtual node's addition has a significant impact on the quality of the node embeddings for some of the algorithms on the *ogbg-molhiv* dataset. We have tested our node embedding with and without the virtual node and observed considerable performance improvement. We have updated Table 1 in the revised manuscript to include our classification results without the virtual node as well.
>
> ___
>
> **Q10.** Why is the variance of the proposed method high for some of the data sets of the 'TUD' repository?
>
> **R.** We believe that this is the consequence of the high-dimensional nature of the embedding. More precisely, for a graph with $N_i$ nodes and the node embedding $Z_i\in\mathbb{R}^{N_i\times d}$, the graph embedding with respect to the reference $Z_0\in \mathbb{R}^{N\times d}$ is $dN$-dimensional. Therefore, for smaller datasets, given the high-dimensionality of the embedding, the variance could be high.

---

> > ### Comment · AnonReviewer1 · 2020-11-24
> > **Thanks**
> >
> > Thanks a lot for the detailed comments and feedback, which I appreciate a lot. Upon re-reading, in one of the newly-added sections on p. 4, I found a 'bare' reference to 'Muandet et al. (2017)', which should probably be added using `\citep` or something. I am impressed by the quality of the updates and thank the authors a lot—this is how a rebuttal session should ideally work!

---

> > > ### Author Response · Authors · 2020-11-24
> > > **Reference corrected.**
> > >
> > > Thank you very much for your prompt and positive evaluation of the updates and for pointing out the bare reference on page 4. We have revised the manuscript to correct that.

---

### Official Review · AnonReviewer2 · 2020-10-26
**Promising but can be stenghten**

**Rating:** 7
**Confidence:** 4

**Review:**

##########################################################################

Summary:

This paper proposes to use a Wasserstein embedding to compare graph using also node embedding methods to convert graphs into vectors. While the elements are not new, the proposed method is new and is competitive with other classical methods (graph kernel and graph neural networks).

##########################################################################

Reasons for score:

Overall, I vote for mild acceptation. The method is clearly new, but the ingredients are already new and the results are not godd but not exiting.

##########################################################################

Pros:

1. The combination of node embedding with Wasserstein distance is new and show pretty good results.

2. The method itself seems very flexible and effective compare to other state of art methods.

3. The presentation of the methods is rather clear and most of the technical details are available. As such the paper is self-contained.

##########################################################################

Cons:

1. The framework relies on a node embedding method. Seems the choice of the method could be crucial, we need some discussion on this point.

2. The proposed node embedding formula (equation 9) assumes that the information on the edge is a scalar. However previously in the paper the edge attributes are a vector, please give some slues on how to deal with such edges.

3. The section 4.2 is not clear for me. Is the zero distribution compute on the whole database? Or is it compute class by class? Both ways seems possible please clarify?

##########################################################################

Questions during rebuttal period:

Please address and clarify the cons above.

---

> ### Author Response · Authors · 2020-11-23
> **Response to Reviewer 2**
>
> We would like to thank the reviewer for the constructive points and her or his valuable time. Below please find our answers.
>
> ---
>
> **Q1.** The framework relies on a node embedding method. It seems the choice of the method could be crucial; we need some discussion on this point.
>
> **R.** We agree with the reviewer in that the choice of the node embedding matters. To keep our experiments similar to that of Togninalli et al. 2019, we used a diffusion-based node embedding. However, there remain many other choices that one can leverage, and maybe more importantly, the embedding could be learned together with the Wasserstein embedding (we added a discussion on this point).
>
> ---
> **Q2.** The proposed node embedding formula (equation 9) assumes that the information on the edge is a scalar. However, previously in the paper, the edge attributes are a vector, please give some clues on how to deal with such edges.
>
> **R.** As mentioned in Appendix A.2, for the cases with multiple edge attributes (e.g., in *PTC-MR* and *ogbg-molhiv* datasets), we use an extension of equation 9 in the paper, where the underlying graph with multi-dimensional edge features is broken into multiple parallel graphs with non-negative single-dimensional edge features, and the parallel graphs perform message passing at each round/layer of the diffusion process. In particular, we first convert the edge feature vectors to binary feature vectors (e.g., using one-hot encoding). Then, using $w_{uv}\in \\{0,1\\}^E$ to denote the $E$-dimensional binary feature vector of any edge $(u,v)\in\mathcal{E}$, we modify equation 9 as
>
> $x_v^{(l)}=\sum_{u\in\mathcal{V}} \left(\sum_{e=1}^E \frac{w_{uv,e}}{\sqrt{\mathsf{deg}_e(u)\mathsf{deg}_e(v)}}\right) x_u^{(l-1)},$
>
> $\forall l\in\\{1,\dots,L\\},  \forall v\in\mathcal{V},$ where for any $e\in\\{1,\dots,E\\}$, $w_{uv,e}$ denotes the $e^{th}$ element of $w_{uv}$, and for any node $v\in\mathcal{V}$, we define  $\mathsf{deg}_e(v)$ as its degree over the $e^{th}$ elements of the edge features; i.e.,
>
> $\mathsf{deg}_{e}(v)=\\sum_\{u \in \mathcal{V} \}w_{uv,e}.$
>
> We assign vectors of all-one features to the self-connections in the graph; i.e., $w_{vv,e}=1,~\forall v\in\mathcal{V}, \forall e\in\\{1,\dots,E\\}$.
>
> ---
> **Q3.** The section 4.2 is not clear for me. Is the zero distribution compute on the whole database? Or is it compute class by class? Both ways seems possible please clarify?
>
> **R.** In our paper, the reference is calculated based on the entire training dataset. This leads to a single embedding (i.e., the tangent space at the reference) for the entire dataset. However, your point is well taken that one can define multiple references that would lead to class-conditional embeddings. A similar idea of having multiple Wasserstein embeddings with different references was proposed in ([Park and Thorpe, CVPR 2018](https://openaccess.thecvf.com/content_cvpr_2018/html/Park_Representing_and_Learning_CVPR_2018_paper.html)). For simplicity, we decided to go with a single embedding to avoid a convoluted method.

---

### Official Review · AnonReviewer3 · 2020-10-28
**A simple but interesting combination of 'linear' OT and graph embeddings, with less than convincing evaluation**

**Rating:** 6
**Confidence:** 4

**Review:**

Strengths:
* **Computational advantage**. The quadratic-to-linear complexity reduction for pairwise computation (each of them an LP) of LOT is clearly the main selling point of this paper, which makes it an appealing approach for comparing multiple graphs.
* **Writing**. The paper is overall very clear, well-written, and engaging.

Weaknesses:
* **Novelty**. The paper proposes a farily trivial combination of Wang et al's 'linear' OT with simple message passing on graphs. The main aspect where there could have been an interesting novel contribution, the computation of the reference embedding, is solved somewhat unsatisfactorily with a k-means embedding
* **Unexplained design choices**. The use of the 'virtual node' is not well motivated, nor its contribution evaluatued experimentally. In addition, it's not clear why it was decided to not implement/comparse the entropy-regularized version of the proposed method, given that this could hold the key to truly scalable graph comparison. In addition, entropy regularization has consistently shown to even *improve* performance compared to exact OT in many applications. Most OT libraries already include this variant, so there really seems to be no excuse not to incoporate it.
* **Some important trade-offs brushed under the rug**. Equation (13) seems to suggest that a NxNxN tensor has to be stored in memory, which could be prohibitive in many settings. Is it fair to say that LOT is implicitly trading time by space complexity? In fact, one could argue that this not even entirely correct, as the LP solved in (13) is now cubic rather than quadratic on N, so it seems there is a trade-off in computation too: reducing the number of pairwise comparisons but making each individual LP problem larger.
* **Experimental results raise questions**. There are many moving parts in the evaluation (e.g., the addition of the virtual node, different types of classifiers), so it's hard to disentangle the core effect of the proposed method.
    * The results in Table 1 seem to indicate that the classifier at the end of the pipeline has a much stronger effect on performance than the proposed method.
    * What kind of classifiers are used in the other GNN methods? Where these also optimized for performance? Why not using the same type of classifier on all methods for fair comparison?
    * All the baseline methods have a very strong degradation in performance between validation and test sets. For the proposed method, the degradation is much less signigicant, especially for the +AutoML classifier. How do the authors explain this? This seems to indicate that there is significant overfitting in all methods, and that proper regularization of the classifiers (e.g., via AutoML) is mitigating this for WEGL. This casts significant doubts on the results of this section in my opinion.
    * Again, for the results in Section 2, the classifier seems to have an important effect in the performance of the proposed methods, yet there is no discussion on the classifiers used in all the baselines. In addition, the standard errors are so large that pretty much all methods have overlapping confiedence intervals, so it's hard to draw a statistically significant conclusion from these results.
    * The runtime comparison is also limited. First, it's not clear whether the runtimes shown for WEGL include the full pipeline (e.g., including the reference embedding computation). Second, one of the methods is run on GPU while the other two are run on CPU, so it's hard to put those two sets of results in direct comparison.

* **Misses importart related work**. E.g.,
    * [1] attempts to realize Characteristic (4) mentioned in Section 3
    * [2] also leverages generalized geodesic for comparison of multiple measures
    * [3] combines OT and WL Kernels and applies it to settings very similar to the ones tackled here.

Other comments questions:
* It would be informative to see a discussion on how well LOT enfoces Characteristic 4, that is, what guarantees does one have on how close the l2 embedded distances is to the true wasserstein distance?

Missing references:
* [1] Courty et al., "Learning Wasserstein Embeddings", 2017
* [2] Seguy and Cuturi, "Principal Geodesic Analysis for Probability Measures under the Optimal Transport Metric", NeurIPS 2015.
* [3] Bécigneul et al., "Optimal Transport Graph Neural Networks", 2020

---

> ### Author Response · Authors · 2020-11-23
> **Response to Reviewer 3 (Part 1)**
>
> We genuinely thank the reviewer for the feedback and the careful evaluation of our work. We have taken into account your feedback, and below, please find our answers to your specific questions and concerns.
>
> ---
>
> **Q1.** The computation of the reference embedding is solved somewhat unsatisfactorily with a k-means embedding.
>
> **R.** We agree with the reviewer that one can take more interesting approaches towards calculating the reference measure. Some of these approaches include calculating the Wasserstein barycenter (could be computationally expensive) or concurrent training of the reference measure. In this paper, we decided to go for simplicity and practicality to avoid unnecessary complexity.
>
> ---
>
> **Q2.** Why don't you use the entropy-regularized version of Kantorovich's problem?
>
> **R.** This is a good point; we have no prejudice against the use of the entropy-regularized optimal transport, and, in fact,  we say in our paper that using the entropy regularized version of the Kantorovich problem would improve our computational performance for datasets containing graphs with a large number of nodes. However, for the experiments presented in the paper, the average number of nodes for the majority of datasets is smaller than 250 (with the majority being smaller than 50). In these settings, linear programming is arguably more efficient than the Sinkhorn algorithm (due to the iterative nature of the Sinkhorn algorithm). To show this, we have added a section to our supplementary material that compares the performance of the linear programming (LP) solver with the Sinkhorn algorithm (on entropy regularized Kantorovich problem) on the *ogbg-molhiv* and the *ogbg-ppa* datasets for various regularization parameters. Finally, while there might be faster solvers (as we point out in the paper), given the acceptable performance of the linear programming solver (at least for the graph datasets in this paper), we used the simpler solver for demonstration purposes.
>
> ---
> **Q3** Does LOT require solving a larger LP problem (i.e., with a cubic transport plan)? Is it trading time by space complexity? Are you "brushing (details) under the rug"?
>
> **R.** We thank the reviewer for reading the details in our supplementary material. However, we do not appreciate using the term "brushed under the rug" by the reviewer, as it goes against who we are and our genuine efforts in writing a transparent paper. Moreover, the raised point is false, and due to a misunderstanding.
>
> We do not require storing or solving for a $N\times N\times N$ tensor at any step in our algorithm. Equation (13) is meant to provide a geometric meaning of the LOT distance from an optimal transportation point of view. It further provides insights into the meaning of the approximation of the Wasserstein distance in the tangent space that we talk about in the paper (see Figure 4 (a) and (b)). In particular,  the barycentric projection, i.e., the transition from Figure 4 (a) to (b), ameliorates the need for having a $N\times N\times N$ transportation plan.
>
> Let us further elaborate on this; embedding each distribution into the tangent space requires solving a classic LP problem between samples of $\mu_i$, $Z_i\in\mathbb{R}^{N_i\times d}$, and fixed samples of the reference $\mu_0$, $Z_0\in\mathbb{R}^{N_0\times d}$, (which is quadratic in $N$, i.e., $N_0\times N$). The embedding $\phi(Z_i)\in \mathbb{R}^{N\times d}$ is obtained by barycentric projection of the optimal transport plan (See Figure 4 (b)). Then the distance between $\mu_i$ and $\mu_j$ is calculated simply as the Euclidean distance between the embedded points $\phi(Z_i)$ and $\phi(Z_j)$. The trade-off we pay in using the LOT framework is that we are not calculating the Wasserstein distance, but really the LOT distance shown in Figure (4), which approximates the Wasserstein distance (See our response to Reviewer 1 for the fidelity of this approximation).
>
> ---
>
> **Q4.** The results indicate that the classifier has a much stronger effect on performance than the proposed method.
>
> **R.** The choice of the classifier indeed has a large effect on the performance. However, we respectfully point out that our proposed fixed embedding allows us to try different classifiers. To exclusively show the effect of the linear Wasserstein embedding, we provided the following ablation study that demonstrates the power of our proposed graph embedding.
>
> For the experiment on the *ogbg-molhiv* dataset, for the same node embedding, we substituted our Wasserstein embedding with a Global Average Pooling (GAP) to obtain the graph embedding. Then we compared the performance of classifiers on the GAP embedded graph and WEGL and show that WEGL provides significantly higher performance.
>
> ---

---

> ### Author Response · Authors · 2020-11-23
> **Response to Reviewer 3 (Part 2)**
>
> **Q5.** Why do all baselines suffer from overfitting? Why is overfitting less in WEGL?
>
> **R.** Over-fitting and over-smoothing are known as the main obstacles of developing reliable deep GNNs, and there is a large body of recent work devoted to this topic. WEGL suffers less from overfitting because 1) the node embedding is fixed and not learnable, and 2) the Wasserstein embedding has been shown to convexify sets of probability measures, hence allowing for efficient modeling of distributions (Aldroubi et al. 2020).
>
> ---
> **Q6.** Why not using the same type of classifier on all methods for fair comparison?
>
> **R.** First, we note that differentiation between GNN and graph kernels is crucial here. GNNs rely on a pooling layer to obtain a fixed-length representation of the entire graph (from latent node representations). Generally speaking, after such a pooling, GNNs apply a softmax classifier (or several layers of fully-connected layers followed by a softmax classifier for obtaining a nonlinear classifier) to classify the graph. All the parameters are then learned in an end-to-end manner.
>
> On the other hand, graph kernels define a (dis)similarity between two graphs and use classic kernel methods (e.g., kernel-SVM) to classify the input graph. These methods do not necessarily provide a fixed length embedding for a graph, and hence would be limited to the kernel methods.
>
> Now, one cannot simply use the classifiers used in our framework on GNNs or graph kernels. That said, there is a direct relationship between the WWL framework, which is a graph kernel method that uses kernel-SVM with (dis)similarity calculated by the Wasserstein distance, i.e.,  $k(G_i,G_j)=\exp(-\alpha W^2_2(\mu_i,\mu_j))$, and applying a kernel-SVM with an RBF kernel in our framework, i.e., $k(G_i,G_j)=\exp(-\alpha \\|\phi(Z_i)-\phi(Z_j)\\|_2^2)$, as $\\|\phi(Z_i)-\phi(Z_j)\\|_2^2\approx W^2_2(Z_i,Z_j)$.
>
> ---
>
> **Q7.** The runtime comparison is limited, as GPU and CPU computations are mixed. Are they calculated for the entire pipeline?
>
> **R.** The runtimes are calculated for the entire pipeline for all the methods, to provide a fair comparison. The modern deep GNNs are too expensive to train on CPUs, and hence the choice of using GPUs. The choice of GPU and CPU calculations by itself shows the training speed up obtained via WEGL compared to a GNN. However, out of respect for the reviewer, we have added the runtimes of GIN on the CPU to Figure 3. Not surprisingly, both training and inference are significantly slower on CPU for almost all datasets, making our method the best choice for both training and evaluation on CPU.
>
> ---
> **Q8.** It would be informative to see a discussion on how well LOT enforces Characteristic 4, that is, what guarantees does one have on how close the l2 embedded distances are to the true Wasserstein distance?
>
> **R.** This is a good point that is also raised by Reviewer 1. Please see our in-depth answer to Reviewer 1 (Q5) on this topic.
>
> ---
>
> **Q9.** Missing important related work.
>
> **R.** Thank you for pointing out the missing references. We have added the suggested references to our paper. Also, it seems like the suggested reference [3] is an exciting approach that is concurrently being reviewed at ICLR 2021.  Bécigneul et al. [3] utilize the idea of learning node embedding prototypes with the Wasserstein distance (i.e., Wasserstein barycenters). Also, the idea overlaps with the CVPR 2020 work by Zhang et al. ["Deep EMD: ..."](https://openaccess.thecvf.com/content_CVPR_2020/html/Zhang_DeepEMD_Few-Shot_Image_Classification_With_Differentiable_Earth_Movers_Distance_and_CVPR_2020_paper.html). We added a few words on these approaches to the paper.
>
> ---
>
> **Q10.** The use of the 'virtual node' is not well motivated, nor its contribution evaluated experimentally.
>
> **R.** The addition of the virtual node simplifies message passing among nodes in datasets whose graphs resemble a tree-like structure, such as in the *ogbg-molhiv* dataset. According to the [official OGB leaderboard](https://ogb.stanford.edu/docs/graphprop/#ogbg-mol), the virtual node's addition has a significant impact on the quality of the node embeddings for some of the algorithms on the *ogbg-molhiv* dataset. We have tested our node embedding with and without the virtual node and observed considerable performance improvement. We have updated Table 1 in the revised manuscript to include our classification results without the virtual node as well.

---

### Official Review · AnonReviewer4 · 2020-10-28
**The paper has merits, but I have a few concerns about the experimental analysis.**

**Rating:** 6
**Confidence:** 3

**Review:**


The paper presents an embedding method for graphs, denoted as WEGL. The method operates by first representing each graph as a set of vector representations of its nodes, secondly, an optimal transport map is computed with respect to a finite reference set of points and, finally, a fixed-size vector representation is computed from that map.
The result is an embedding into a Hilbert space, where the distance between two graph representations approximates the 2-Wasserstein distance between their respective node distributions.

The paper is partly inspired by WWL (Togninalli et al. 2019) and provides an explicit embedding map, hence improving in terms of computational performance. I have some comments and concerns about the experimental analysis, which does not seem to directly validate the paper claims. Overall, it is well written and clear in almost all its parts.

Comments.
1. The paper considers only the smallest of the datasets in OGB (Hu et al., 2020) for graph-level prediction, specifically ogbg-molhiv. I wonder if the proposed method, which is claimed to be computationally efficient, can handle also the other three.
2. Choice of the baseline methods. I expected comparisons with other (explicit) embedding methods (eg, Hu* et al. 2020, Kriege et al. 2014) both computationally and at the task. Since WEGL is somehow a variation of WWL, I would like to see its performance compared also in Section 5.1.
3. How performance has been assessed. The paper mention in the captions of Tables 1 and 2 that some of the results have been reported from already published papers. However, I couldn't find any details about the experimental settings of the other methods. Is the architecture of the GIN in Figure 3 the same as that in Tables 1 and 2? Are all the baseline methods operating on graphs with the "virtual node" variant (which is important for a fair comparison of the embedding capability)?

To further improve the paper, I suggest spending a few more words to
- clarify the use of the Jacobian and the approximation in points 1 and 3, Section 3.1;
- define the barycentric project and clarify why it is not invertible.

(Kriege et al. 2014) Kriege, Nils, et al. "Explicit versus implicit graph feature maps: A computational phase transition for walk kernels." 2014 IEEE international conference on data mining. IEEE, 2014.

---

> ### Author Response · Authors · 2020-11-23
> **Response to Reviewer 4**
>
> We thank the reviewer for the feedback and for her or his invaluable time.
>
> ---
>
> **Q1.** Could the method be applied to larger datasets?
>
> **R.** We started experimenting with larger graph property prediction datasets in the OGB database, including the *ogbg-molpcba* and *ogbg-ppa* datasets. Unfortunately, we were not able to finish the experiments during the rebuttal period. We will continue working on both datasets, and we hope to add graph classification results on them to the camera-ready version of the paper.
>
> ---
> **Q2.** Baselines and adding WWL in Section 5.1.
>
> **R.** Regarding the comment about adding WWL results on *ogbg-molhiv*, we could not afford this computation during the rebuttal period. More precisely, the training set contains $32900$ graphs, and to calculate the similarity kernel matrix required for WWL, one needs to calculate more than half a billion linear programming problems (or the entropic regularized version of it) corresponding to the Wasserstein distances between the graph pairs, i.e.,  ${32900 \choose 2} =541,188,550$. On our machines, each LP problem takes on average about $1.0 ms$, which translates to about $150 hrs$ of computations for the training split only. An additional $75 hrs$ is needed for calculating the similarity kernel for the test and validation splits, which sums to $225 hrs$. Hence, we could not finish the experiment during the rebuttal period. For larger datasets, such as *ogbg-ppa*, WWL's calculation is too expensive and only feasible if one has access to many CPU cores.
>
> ---
> **Q3** How was the performance assessed in Tables 1 and 2? Did all methods use a "virtual node?"
>
> **R.** In each of the methods, both proposed by us and the baselines from the literature, the results are reported based on the best set of hyperparameters, either evaluated on the validation set or during cross-validation if the dataset does not provide a separate validation split. Moreover, for run-time comparison in Figure 3, we considered a similar structure for GIN to that of the original GIN paper with 5 hidden layers. We added this point to the caption of Figure 3 in the revised manuscript. Finally, for the baseline results in Table 1 on the *ogbg-molhiv* dataset, we included the best results reported in each corresponding paper and the [official online leaderboard](https://ogb.stanford.edu/docs/graphprop/#ogbg-mol), where some of the methods (e.g., GIN) use a virtual-node augmentation of the dataset while the rest do not. We have updated Table 1 in the revised manuscript to include our classification results without the virtual node as well.
>
> ---
>
> **Q4.** clarification on the use of the Jacobian and the approximation and missing reference.
>
> **R.** We have added more details on the role of the Jacobian equation, the push forward and backward of a measure, and further elaborated about the embedding properties for the audience who might be less familiar with the optimal transport problem. Finally, we added the missing reference and revised the paper accordingly.

---

### Author Response · Authors · 2020-11-23
**Summary of Changes**

We sincerely thank the reviewers for their effort and their helpful comments regarding our paper. We have carefully revised the manuscript according to your comments. A summary of the main changes we have made are as follows:

* We have streamlined the paper's structure by moving the less relevant material, such as the inner working of GNNs, to the appendix.
* We have added details on the approximation error and the regularity properties of the proposed embedding.
* We have improved our experimental results on the *ogbg-molhiv* dataset by including:
    * Results using WEGL and a random forest classifier without leveraging a virtual node variant of the graphs; and,
    * An ablation study on replacing the proposed Wasserstein embedding module with global average pooling while using the same node embedding process and downstream classifier (i.e., random forest) as our method.
* We have added discussions on the effectiveness of the entropy-regularized solver and when to leverage it.
* We have included run-time results for the GIN baseline using CPU hardware.
* We have added several missing references from the related work in the literature pointed out by the reviewers.

We hope that our revised manuscript and our responses fully address your comments. Thank you again for selflessly offering your invaluable time to the community.

---

### Decision · Program_Chairs · 2021-01-07
**Final Decision**

**Decision:**

Accept (Poster)

**Comment:**

This paper proposes  a novel and interesting embedding of graphs emulating the Wasserstein distance. The experiments are good and the authors did a detailed answer taking into account the comments of the reviewer. The responses were appreciated and the AC recommends the paper to be accepted.